# Which Models have Perceptually-Aligned Gradients?
# An Explanation via Off-Manifold Robustness

**Suraj Srinivas**[*]
Harvard University
Cambridge, MA
ssrinivas@seas.harvard.edu

**Sebastian Bordt**[*]
University of Tübingen, Tübingen AI Center
Tübingen, Germany
sebastian.bordt@uni-tuebingen.de

**Himabindu Lakkaraju**
Harvard University
Cambridge, MA
hlakkaraju@hbs.edu

## Abstract

One of the remarkable properties of robust computer vision models is that their input-gradients are often aligned with human perception, referred to in the literature as perceptually-aligned gradients (PAGs). Despite only being trained for classification, PAGs cause robust models to have rudimentary generative capabilities, including image generation, denoising, and in-painting. However, the underlying mechanisms behind these phenomena remain unknown. In this work, we provide a first explanation of PAGs via *off-manifold robustness*, which states that models must be more robust off- the data manifold than they are on-manifold. We first demonstrate theoretically that off-manifold robustness leads input gradients to lie approximately on the data manifold, explaining their perceptual alignment. We then show that Bayes optimal models satisfy off-manifold robustness, and confirm the same empirically for robust models trained via gradient norm regularization, randomized smoothing, and adversarial training with projected gradient descent. Quantifying the perceptual alignment of model gradients via their similarity with the gradients of generative models, we show that off-manifold robustness correlates well with perceptual alignment. Finally, based on the levels of on- and off-manifold robustness, we identify three different regimes of robustness that affect both perceptual alignment and model accuracy: weak robustness, bayes-aligned robustness, and excessive robustness. Code is available at https://github.com/tml-tuebingen/pags.

## 1 Introduction

An important desideratum for machine learning models is *robustness*, which requires that models be insensitive to small amounts of noise added to the input. In particular, *adversarial robustness* requires models to be insensitive to adversarially chosen perturbations of the input. Tsipras et al. [1] first observed an unexpected benefit of such models, namely that their input-gradients were "significantly more human-aligned" (see Figure 1 for examples of perceptually-aligned gradients). Santurkar et al. [2] built on this observation to show that robust models could be used to perform rudimentary image synthesis - an unexpected capability of models trained in a purely discriminative manner. Subsequent works have made use of the perceptual alignment of robust model gradients to improve zero-shot

---

[*]Equal Contribution

37th Conference on Neural Information Processing Systems (NeurIPS 2023).

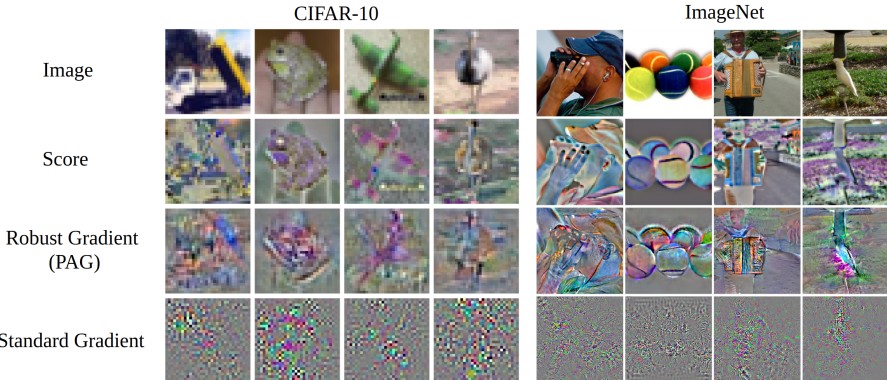

Figure 1: A demonstration of the perceptual alignment phenomenon. The input-gradients of robust classifiers ("robust gradient") are perceptually similar to the score of diffusion models [10], while being qualitatively distinct from input-gradients of standard models ("standard gradient"). Best viewed in digital format.

object localization [3], perform conditional image synthesis [4], and improve classifier robustness [5, 6]. Kaur et al. [7] coined the term *perceptually-aligned gradients* (PAGs), and showed that it occurs not just with adversarial training [8], but also with randomized smoothed models [9]. Recently, Ganz et al. [6] showed that the relationship between robustness and perceptual alignment can also work in the opposite direction: approximately enforcing perceptual alignment of input gradients can increase model robustness.

Despite these advances, the underlying mechanisms behind the phenomenon of perceptually aligned gradients in robust models are still unclear. Adding to the confusion, prior works have used the same term, PAGs, to refer to slightly different phenomena. We ground our discussion by first identifying variations of the same underlying phenomenon.

**Phenomenon 1** (Perceptual Alignment). *The gradients of robust models highlight perceptually relevant features [1], and highlight discriminative input regions while ignoring distractors [11].*

**Phenomenon 2** (Generative Capabilities). *Robust models have rudimentary image synthesis capabilities, where samples are generated by maximizing a specific output class by iteratively following the direction dictated by the input-gradients [2].*

**Phenomenon 3** (Smoothgrad Interpretability). *Smoothgrad visualizations (which are gradients of randomized smooth models) tend to be visually sharper than the gradients of standard models [12].*

While Kaur et al. [7] refer to Phenomenon 2 as PAGs, Ganz et al. [6] use this term to refer to Phenomenon 1. Since they are all related to the gradients of robust models, we choose to collectively refer to all of these phenomena as PAGs.

While the various phenomena arising from PAGs are now well-documented, there is little to no work that attempts to explain the underlying mechanism. Progress on this problem has been hard to achieve because PAGs have been described via purely qualitative criteria, and it has been unclear how to make these statements quantitative. In this work, we address these gaps and make one of the first attempts at explaining the mechanisms behind PAGs. Crucially, we ground the discussion by attributing PAGs to gradients lying on a manifold, based on an analysis of Bayes optimal classifiers. We make the following contributions:

1. We establish the first-known theoretical connections between Bayes optimal predictors and the perceptual alignment of classifiers via *off-manifold* robustness. We also identify the manifold w.r.t. which this holds, calling it the *signal manifold*.

2. We experimentally verify that models trained with gradient-norm regularization, noise augmentation and randomized smoothing all exhibit off-manifold robustness.

3. We *quantify* the perceptual alignment of gradients by their perceptual similarity with the score of the data distribution, and find that this measure correlates well with off-manifold robustness.

4. We identify **three regimes of robustness:** *weak* robustness, *bayes-aligned* robustness and *excessive* robustness, that differently affect both perceptual alignment and model accuracy.

## 2 Related Work

**Robust training of neural networks**   Prior works have considered two broad classes of model robustness: adversarial robustness and robustness to normal noise. Adversarial robustness is achieved by training with adversarial perturbations generated using project gradient descent [8], or by randomized smoothing [9], which achieves certified robustness to adversarial attacks by locally averaging with normal noise. Robustness to normal noise is achieved by explicitly training with noise, a technique that is equivalent to Tikhonov regularization for linear models [13]. For non-linear models, gradient-norm regularization [14] is equivalent to training with normal noise under the limit of training with infinitely many noise samples [15]. In this paper, we make use of all of these robust training approaches and investigate their relationship to PAGs.

**Gradient-based model explanations**   Several popular post hoc explanation methods[16, 17, 12] estimate feature importances by computing gradients of the output with respect to input features and aggregating them over local neighborhoods [18]. However, the visual quality criterion used to evaluate these explanations has given rise to methods that produce visually striking attribution maps, while being independent of model behavior [19]. While [20] attribute visual quality to implicit score-based generative modeling of the data distribution, [21] propose that it depends on explanations lying on the data manifold. While prior works have attributed visual quality to generative modeling of data distribution or explanations lying on data manifold, our work demonstrates for the first time that (1) off-manifold robustness is a crucial factor, and (2) it is not the data manifold / distribution, rather the signal manifold / distribution (defined in Section 3.2) that is the critical factor in explaining the phenomenon of PAGs.

## 3 Explaining Perceptually-Aligned Gradients

Our goal in this section is to understand the mechanisms behind PAGs. We first consider PAGs as lying on a low-dimensional manifold, and show theoretically that such on-manifold alignment of gradients is equivalent to off-manifold robustness of the model. We then argue that Bayes optimal models achieve both off-manifold robustness and on-manifold gradient alignment. In doing so, we introduce the distinction between the data and the signal manifold, which is key to understanding the manifold structure of PAGs. Finally, we present arguments for why empirical robust models also exhibit off-manifold robustness.

**Notation**   Throughout this paper, we consider the task of image classification with inputs $\mathbf{x} \in \mathbb{R}^d$ where $\mathbf{x} \sim \mathcal{X}$ and $y \in [1, 2, ...C]$ with $C$-classes. We consider deep neural networks $f : \mathbb{R}^d \to \triangle^{C-1}$ which map inputs $\mathbf{x}$ onto a $C$-class probability simplex. We assume that the input data $\mathbf{x}$ lies on a $k$-dimensional manifold in the $d$-dimensional ambient space. Formally, a $k$-dimensional differential manifold $\mathcal{M} \subset \mathbb{R}^d$ is locally Euclidean in $\mathbb{R}^k$. At every point $\mathbf{x} \in \mathcal{M}$, we can define a projection matrix $\mathbb{P}_x \in \mathbb{R}^{d \times d}$ that projects points onto the $k$-dimensional tangent space at $\mathbf{x}$. We denote $\mathbb{P}_x^{\perp} = \mathbb{I} - \mathbb{P}_x$ as the projection matrix to the subspace orthogonal to the tangent space.

### 3.1 Off-Manifold Robustness ⇔ On-Manifold Gradient Alignment

We now show via geometric arguments that off-manifold robustness of models and on-manifold alignment of gradients are identical. We begin by discussing definitions of on- and off-manifold noise. Consider a point $\mathbf{x}$ on manifold $\mathcal{M}$, and a noise vector $\mathbf{u}$. Then, $\mathbf{u}$ is **off-manifold** noise, if $\mathbb{P}_x(\mathbf{x} + \mathbf{u}) = \mathbf{x}$, which we denote $\mathbf{u} := \mathbf{u}_{\text{off}}$, and $\mathbf{u}$ is **on-manifold** noise, if $\mathbb{P}_x(\mathbf{x} + \mathbf{u}) = \mathbf{x} + \mathbf{u}$, which we denote $\mathbf{u} := \mathbf{u}_{\text{on}}$. In other words, if the noise vector lies on the tangent space then it is on-manifold, otherwise it is off-manifold. Given this definition, we can define relative off-manifold robustness. For simplicity, we consider a scalar valued function, which can correspond to one of the $C$ output classes of the model.

**Definition 1.** *(Relative off-manifold robustness) A model $f : \mathbb{R}^d \to \mathbb{R}$ is $\rho_1$-off-manifold robust, wrt some noise distribution $\mathbf{u} \sim U$ if*

$$\frac{\mathbb{E}_{\mathbf{u}_{off}} \left( f(\mathbf{x} + \mathbf{u}_{off}) - f(\mathbf{x}) \right)^2}{\mathbb{E}_{\mathbf{u}} (f(\mathbf{x} + \mathbf{u}) - f(\mathbf{x}))^2} \leq \rho_1 \quad where \quad \mathbf{u}_{off} = \mathbb{P}_x^{\perp}(\mathbf{u})$$

While this definition states that the model is more robust off-manifold than it is overall, it can equivalently be interpreted as being more robust off-manifold than on-manifold (with a factor $\frac{\rho_1}{1-\rho_1}$). Let us now define on-manifold gradient alignment.

**Definition 2.** *(On-manifold gradient alignment) A model $f : \mathbb{R}^d \to \mathbb{R}$ has $\rho_2$-on-manifold aligned gradients if*

$$\frac{\|\nabla_{\mathbf{x}} f(\mathbf{x}) - \nabla_{\mathbf{x}}^{on} f(\mathbf{x})\|^2}{\|\nabla_{\mathbf{x}} f(\mathbf{x})\|^2} \leq \rho_2 \quad where \quad \nabla_{\mathbf{x}}^{on} f(\mathbf{x}) = \mathbb{P}_x \nabla_{\mathbf{x}} f(\mathbf{x})$$

This definition captures the idea that the difference between the gradients and its on-manifold projection is small. If $\rho_2 = 0$, then the gradients exactly lie on the tangent space. We are now ready to state the following theorem.

**Theorem 1** (Equivalence between off-manifold robustness and on-manifold alignment). *A function $f : \mathbb{R}^d \to \mathbb{R}$ exhibits on-manifold gradient alignment if and only if it is off-manifold robust wrt normal noise $\mathbf{u} \sim \mathcal{N}(0, \sigma^2)$ for $\sigma \to 0$ (with $\rho_1 = \rho_2$).*

*Proof Idea.* Using a Taylor series expansion, we can re-write the off-manifold robustness objective in terms of gradients. We further use linear algebraic identities to simplify that yield the on-manifold gradient alignment objective. The full proof is given in the supplementary material. □

When the noise is small, on-manifold gradient alignment and off-manifold robustness are identical. To extend this to larger noise levels, we would need to make assumptions regarding the curvature of the data manifold. Broadly speaking, the less curved the underlying manifold is (the closer it is to being linear), the larger the off-manifold noise we can add, without it intersecting at another point on the manifold. In practice, we expect image manifolds to be quite smooth, indicating relative off-manifold robustness to larger noise levels [22]. In this next part, we will specify the properties of the specific manifold w.r.t. which these notions hold.

### 3.2 Connecting Bayes Optimal Classifiers and Off-Manifold Robustness

In this subsection, we aim to understand the manifold structure characterizing the gradients of Bayes optimal models. We proceed by recalling the concept of Bayes optimality.

**Bayes optimal classifiers.** If we perfectly knew the data generating distributions for all the classes, i.e., $p(\mathbf{x} \mid y = i)$ for all $C$ classes, we could write the Bayes optimal classifier as $p(y = i \mid \mathbf{x}) = p(\mathbf{x} \mid y = i)/\sum_{j=1}^{C} p(\mathbf{x} \mid y = j)$. This is an oracle classifier that represents the "best possible" model one can create from data with perfect knowledge of the data generating process. Given our assumption that the data lies on a low-dimensional data manifold, the Bayes optimal classifier is *uniquely defined on the data manifold.* However, outside this manifold, its behaviour is undefined as all of the class-conditional probabilities $p(\mathbf{x} \mid y)$ are zero or undefined themselves. In order to link Bayes-optimality and robustness, which is inherently about the behavior of the classifier outside the data manifold, we introduce a ground-truth *perturbed data distribution* that is also defined outside the data manifold. While we might consider many perturbed distributions, it is convenient to consider the data generating distributions that are represented by denoising auto-encoders with a stochastic decoder (or equivalently, score-based generative models [23]). Here, the stochasticity ensures that the data-generating probabilities are defined everywhere, not just on the data manifold. This approach allows us to estimate the data manifold, assuming that the autoencoder has a bottleneck layer with $k$ features for a $d$-dimensional ambient space [22].

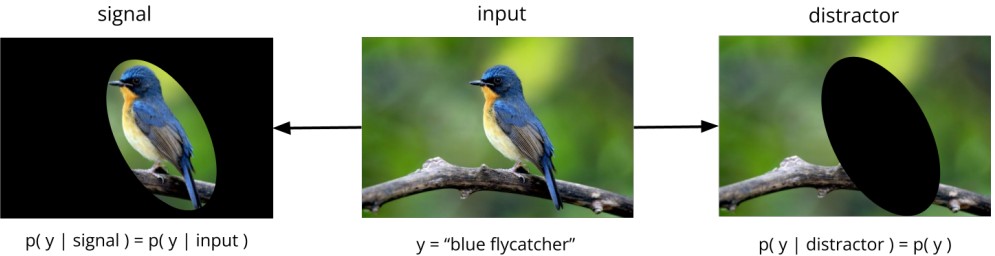

| signal | input | distractor |
|---|---|---|

p( y | signal ) = p( y | input )          y = "blue flycatcher"          p( y | distractor ) = p( y )

Figure 2: An illustration of a signal-distractor decomposition for a bird classification task. The signal represents the discriminative parts of the input, while the distractor represents the non-discriminative parts.

**Distinguishing Between the Data and the Signal Manifold.**   Given a classification problem, one can often decompose the inputs into a *signal* component and a *distractor* component. For intuition, consider the binary task of classifying cats and dogs. Given an oracle data generating distribution of cats and dogs in diverse backgrounds, the label must be statistically independent of the background and depend purely on the object (cat or dog) in the image. In other words, there exist no spurious correlations between the background and the output label. In such a case, we can call the object the signal and the background the distractor. Formally, for every input $\mathbf{x}$ there exists a binary mask $\mathbf{m}(\mathbf{x}) \in \{0,1\}^d$ such that the signal is given by $s(\mathbf{x}) = \mathbf{x} \odot \mathbf{m}(\mathbf{x})$ and the distractor is given by $d(\mathbf{x}) = \mathbf{x} \odot (1 - \mathbf{m}(\mathbf{x}))$. The signal and distractor components are orthogonal to one another $(s(\mathbf{x})^\top d(\mathbf{x}) = 0)$, and we can decompose any input in this manner ($\mathbf{x} = s(\mathbf{x}) + d(\mathbf{x})$). We provide an illustrative figure in Figure 2. Using this, we can now define the signal-distractor distributions.

**Definition 3.** *Given a data distribution $p(\mathbf{x} \mid y)$ for $y \in [1, C]$, we have masking functions $\mathbf{m}(\mathbf{x})$ such that the resulting distribution $p(\mathbf{x} \odot (1 - \mathbf{m}) \mid y) = p(\mathbf{x} \odot (1 - \mathbf{m}))$ is statistically independent of $y$. The sparsest such masking function (such that $\mathbf{m}^* = \arg\min_{\mathbf{m}} \mathbb{E}_{\mathbf{x} \sim p(\mathbf{x}|y)} \|\mathbf{m}(\mathbf{x})\|_0$), yields a corresponding distribution $p(\mathbf{x} \odot (1 - \mathbf{m}^*(\mathbf{x})))$, which is the distractor distribution, and its counterpart $p(\mathbf{x} \odot \mathbf{m}^*(\mathbf{x}) \mid y)$ the signal distribution.*

While it is possible to define signals and distractors more generally as any linear projections of the data, in this work, we restrict ourselves to masking to relate to feature attributions [11]. While the subject of finding such optimal masks is the topic of feature attribution [24], in this discussion, we shall assume that the optimal masks $\mathbf{m}^*$ and the corresponding signal and distractor distributions are known. Similar to decomposing any point on the manifold, we can also decompose any vector on the tangent space on the data manifold into signal and distractor components using the optimal mask $\mathbf{m}^*$. In other words, we can write

$$\nabla_{\mathbf{x}} p(\mathbf{x} \mid y) = \underbrace{\nabla_{\mathbf{x}} p(\mathbf{x} \mid y) \odot \mathbf{m}^*(\mathbf{x})}_{\text{signal } s(\mathbf{x}) \text{ (has information about } y)} + \underbrace{\nabla_{\mathbf{x}} p(\mathbf{x} \mid y) \odot (1 - \mathbf{m}^*(\mathbf{x}))}_{\text{distractor } d(\mathbf{x}) \text{ (independent of y)}}$$

Finally, we note that this signal-distractor decomposition does not meaningfully exist for all classification problems in that it can be trivial with the entire data distribution being equal to the signal, with zero distractors. Two examples of such cases are: (1) ordinary MNIST classification has no distractors due to the simplicity of the task, as the entire digit is predictive of the true label and the background is zero. (2) multi-class classification with a large set of classes with diverse images also have no distractors due to the complexity of the task, as the background can already be correlated with class information. For example, if the dataset mixes natural images with deep space images, there is no single distractor distribution one can find via masking that is independent of the class label. This disadvantage arises partly because we aim to identify important pixels, whereas a more general definition based on identifying subspaces can avoid these issues. We leave the exploration of this more general class of methods to future work.

Given this definition of the signal and distractor distributions, we are ready to make the following theorem: the input-gradients of a Bayes optimal classifier lie on the signal manifold, as opposed to the general data manifold.

**Theorem 2.** *The input-gradients of Bayes optimal classifiers lie on the signal manifold.*

*Proof Idea.* We first show that the gradients of the Bayes optimal predictor lie in the linear span of the gradients of the gradients of the data distribution, thus lying on the tangent space of the data distribution. Upon making a signal-distractor decomposition of the gradients of the Bayes optimal classifier, we find that the distractor term is always zero, indicating that the gradients of the Bayes optimal classifier always lie on the signal manifold. The proof is given in the supplementary material. □

The full proof is given in the Supplementary material. This theorem can be intuitively thought of as follows: the optimal classifier only needs to look at discriminative regions of the input in order to classify optimally. In other words, changing the input values at discriminative signal regions is likely to have a larger effect on model output than changing the inputs slightly at unimportant distractor regions, indicating that the gradients of the Bayes optimal classifier highlight the signal. Thus, the Bayes optimal classifier does not need to model the distractor; only the signal is sufficient. This fact inspires us to make the following hypothesis to help us ground the discussion on the perceptual alignment of gradients:

**Hypothesis 1.** *The input gradients of classifiers are perceptually aligned if and only if they lie on the signal manifold, thus highlighting only discriminative parts of the input.*

This indicates that Bayes optimal model's gradients are perceptually aligned. Recall that the signal component only provides information about discriminative components of the input, and thus, we expect this to connect to perceptual alignment from Phenomenon 1. Next, we ask whether the gradients of practical models, particularly robust models, are also off-manifold robust.

### 3.3 Connecting Robust Models and Off-Manifold Robustness

Previously, we saw that Bayes optimal classifiers have the property of relative off-manifold robustness. Here, we argue that off-manifold robustness also holds for robust models in practice. This is a non-trivial and a perhaps surprising claim: common robustness objectives such as adversarial training, gradient-norm regularization, etc are isotropic, meaning that they do not distinguish between on- and off-manifold directions.

**Hypothesis 2.** *Robust models are off-manifold robust w.r.t. the signal manifold.*

We provide two lines of argument in support for this hypothesis. Ultimately, however, our evidence is empirical and presented in the next section.

**Argument 1: Combined robust objectives are non-isotropic.** While robustness penalties itself are isotropic and do not prefer robustness in any direction, they are combined with the cross-entropy loss on data samples, which lie on the data manifold. Let us consider the example of gradient norm regularization. The objective is given by:

$$\mathbb{E}_{\mathbf{x}}\left(\ell(f(\mathbf{x}), y(\mathbf{x})) + \lambda\|\nabla_{\mathbf{x}}f(\mathbf{x})\|^2\right) = \mathbb{E}_{\mathbf{x}}\left(\underbrace{\ell(f(\mathbf{x}), y(\mathbf{x})) + \lambda\|\nabla_{\mathbf{x}}^{\text{on}}f(\mathbf{x})\|^2}_{\text{on-manifold objective}} + \lambda\underbrace{\|\nabla_{\mathbf{x}}^{\text{off}}f(\mathbf{x})\|^2}_{\text{off-manifold objective}}\right)$$

Here we have used, $\|\nabla_{\mathbf{x}}f(\mathbf{x})\|^2 = \|\nabla_{\mathbf{x}}^{\text{on}}f(\mathbf{x}) + \nabla_{\mathbf{x}}^{\text{off}}f(\mathbf{x})\|^2 = \|\nabla_{\mathbf{x}}^{\text{on}}f(\mathbf{x})\|^2 + \|\nabla_{\mathbf{x}}^{\text{off}}f(\mathbf{x})\|^2$ due to $\nabla_{\mathbf{x}}^{\text{on}}f(\mathbf{x})^{\top}\nabla_{\mathbf{x}}^{\text{off}}f(\mathbf{x}) = 0$, which is possible because the on-manifold and off-manifold parts of the gradient are orthogonal to each other. Assuming that we are able to decompose models into on-manifold and off-manifold parts, these two objectives apply to these decompositions independently. This argument states that in this robust training objective, there exists a trade-off between the cross-entropy loss and the on-manifold robustness term, whereas there is no such trade-off for the off-manifold term, indicating that it is much easier to minimize off-manifold robustness than on-manifold. This argument also makes the prediction that increased on-manifold robustness must be accompanied by higher train loss and decreased out-of-sample performance, and we will test this in the experiments section.

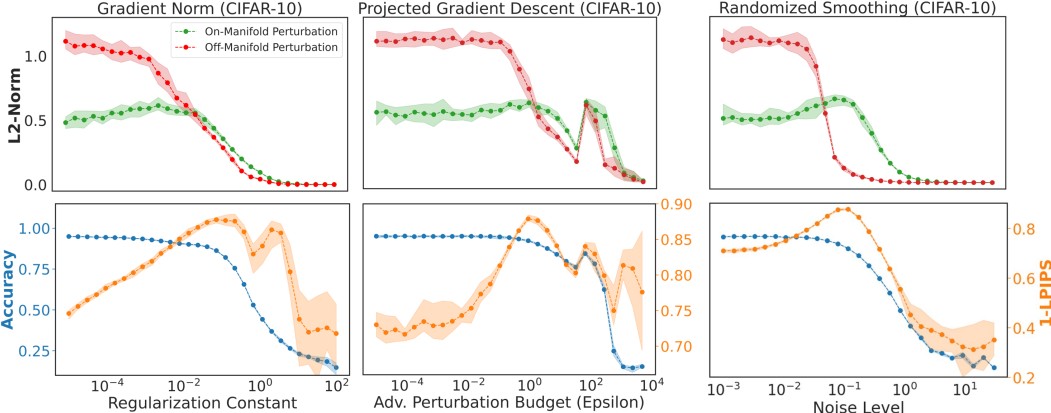

Figure 3: **Top Row:** Robust models are off-manifold robust. The figure depicts the inverse on- and off-manifold robustness of Resnet18 models trained with different objectives on CIFAR-10 (larger values correspond to less robustness). As we increase the importance of the robustness term in the training objective, the models become increasingly robust to off-manifold perturbations. At the same time, their robustness to on-manifold perturbations stays approximately constant. This means that the models become off-manifold robust. As we further increase the degree of robustness, both on- and off-manifold robustness increase. **Bottom Row:** The input gradients of robust models are perceptually similar to the score of the probability distribution, as measured by the LPIPS metric. We can also identify the models that have the most perceptually-aligned gradients (the global maxima of the yellow curves). Figures depict mean and minimum/maximum across 10 different random seeds. Additional results including the smoothness penalty can be found in Supplement Figure 9.

However, there are nuances to be observed here: while the data lies on the data manifold, the gradients of the optimal model lie on the signal manifold so this argument may not be exact. Nonetheless, for cases where the signal manifold and data manifold are identical, this argument holds and can explain a preference for off-manifold robustness over on-manifold robustness.

**Argument 2: Robust linear models are off-manifold robust.** It is a well-known result in machine learning (from, for example the representer theorem) that the linear analogue of gradient norm regularization, i.e., weight decay causes model weights to lie in the linear span of the data. In other words, given a linear model $f(\mathbf{x}) = \mathbf{w}^\top \mathbf{x}$, its input-gradient are the weights $\nabla_\mathbf{x} f(\mathbf{x}) = \mathbf{w}$, and when trained with the objective $\mathcal{L} = \mathbb{E}_\mathbf{x}(f(\mathbf{x}) - y(\mathbf{x}))^2 + \lambda \|\mathbf{w}\|^2$, it follows that the weights have the following property: $\mathbf{w} = \sum_{i=1}^N \alpha_i \mathbf{x}_i$, i.e., the weights lie in the span of the data. In particular, if the data lies on a *linear subspace*, then so do the weights. Robust linear models are also infinitely off-manifold robust: for any perturbation $\mathbf{u}_{\text{off}}$ orthogonal to the data subspace, $\mathbf{w}^\top(\mathbf{x} + \mathbf{u}_{\text{off}}) = \mathbf{w}^\top \mathbf{x}$, thus they are completely robust to off-manifold perturbations.

In addition, if we assume that there are input co-ordinates $\mathbf{x}_i$ that are uncorrelated with the output label, then $\mathbf{w}_i \mathbf{x}_i$ is also uncorrelated with the label. Thus the only way to minimize the mean-squared error is to set $\mathbf{w}_i = 0$ (i.e., a solution which sets $\mathbf{w}_i = 0$ has strictly better mean-squared error than one that doesn't), in which case the weights lie in the signal subspace, which consists of the subspace of all features correlated with the label. This shows that even notions of signal-distractor decomposition transfer to the case of linear models.

## 4 Experimental Evaluation

In this section, we conduct extensive empirical analysis to confirm our theoretical analyses and additional hypotheses. We first demonstrate that robust models exhibit relative off-manifold robustness (Section 4.1). We then show that off-manifold robustness correlates with the perceptual alignment of gradients (Section 4.2). Finally, we show that robust models exhibit a signal-distractor decomposition, that is they are relatively robust to noise on a distractor rather than the signal (Section 4.3). Below we detail our experimental setup. Any additional details can be found in the Supplementary material.

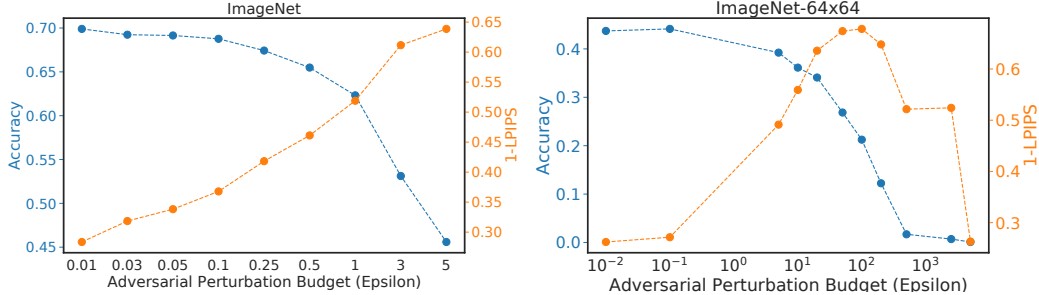

Figure 4: The input gradients of robust models trained with projected gradient descent on ImageNet and Imagenet-64x64 are perceptually similar to the score of the probability distribution, as measured by the LPIPS metric. On Imagenet-64x64, we also trained excessively robust models.

**Data sets and Robust Models.** We use CIFAR-10 [25], ImageNet and ImageNet-64 [26], and an MNIST dataset [27] with a distractor, inspired by [11]. We train robust Resnet18 models [28] with (i) gradient norm regularization, (ii) randomized smoothing, (iii) a smoothness penalty, and (iv) $l_2$-adversarial robust training with projected gradient descent. The respective loss functions are given in the Supplementary material. On ImageNet, we use pre-trained robust models from [29].

**Measuring On- and Off-Manifold Robustness.** We measure on- and off-manifold robustness by perturbing data points with on- and off-manifold noise (Section 3.1). For this, we estimate the tangent space of the data manifold with an auto-encoder, similar to [22, 30, 21]. We then draw a random noise vector and project it onto the tangent space. Perturbation of the input in the tangent direction is used to measure on-manifold robustness. Perturbation of the input in the orthogonal direction is used to measure off-manifold robustness. To measure the change in the output of a classifier with $C$ classes, we compute the $L_2$-norm $||f(x) - f(x + u)||_2$.

**Measuring Perceptual Alignment.** To estimate the perceptual alignment of model gradients, we would ideally compare them with the gradients of the Bayes optimal classifier. Since we do not have access to the Bayes optimal model's gradients $\nabla_{\mathbf{x}} p(y \mid \mathbf{x})$, we use the score $\nabla_{\mathbf{x}} \log p(\mathbf{x} \mid y)$ as a proxy, as both lie on the same data manifold. Given the gradient of the robust model and the score, we use the Learned Perceptual Image Patch Similarity (LPIPS) metric [31] to measure the perceptual similarity between the two. The LPIPS metric computes the similarity between the activations of an AlexNet [32] and has been shown to match human perception well [31]. In order to estimate the score $\nabla_{\mathbf{x}} \log p(\mathbf{x} \mid y)$, we make use of the diffusion-based generative models from Karras et al. [10]. Concretely, if $D(\mathbf{x}, \sigma)$ is a denoiser function for the noisy probability distribution $p(\mathbf{x}, \sigma)$ (compare Section 3.2), then the score is given by $\nabla_{\mathbf{x}} \log p(\mathbf{x}, \sigma) = (D(\mathbf{x}, \sigma) - \mathbf{x})/\sigma^2$ [10, Equation 3]. We use noise levels $\sigma = 0.5$ and $\sigma = 1.2$ on CIFAR-10 and ImageNet-64, respectively.

### 4.1 Evaluating On- *vs.* Off-manifold Robustness of Models

We measure the on- and off-manifold robustness of different Resnet18 models on CIFAR-10, using the procedure described above. We measure how much the model output changes in response to an input perturbation of a fixed size (results for different perturbation sizes can be found in Supplement Figure 9). The models were trained with three different robustness objectives and to various levels of robustness. The results are depicted in the top row of Figure 3. Larger values in the plots correspond to a larger perturbation in the output, that is less robustness. For little to no regularization (the left end of the plots), the models are less robust to random changes off- than to random changes on- the data manifold (the red curves lie above the green curves). Increasing the amount of robustness makes the models increasingly off-manifold robust (the red curves decrease monotonically). At the same time, the robustness objectives do not affect the on-manifold robustness of the model (the green curves stay roughly constant). This means that the robust models become relatively off-manifold robust (Definition 1). At some point, the robustness objectives also start to affect the on-manifold behavior of the model, so that the models become increasingly on-manifold robust (the green curves start to fall). As can be seen by a comparison with the accuracy curves in the bottom row of Figure 3, increasing on-manifold robustness mirrors a steep fall in the accuracy of the trained models (the

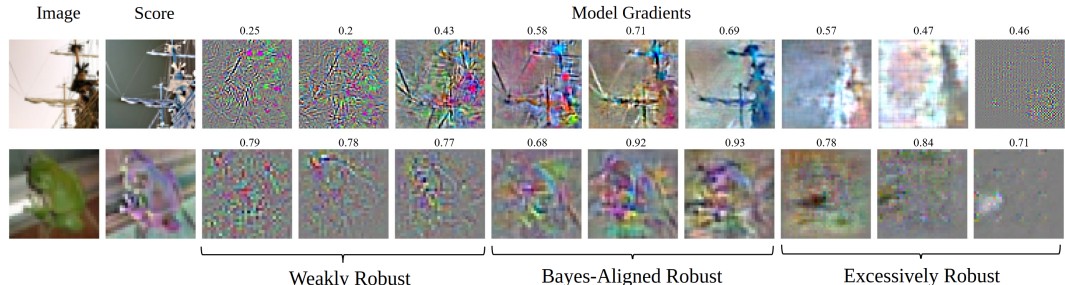

Figure 6: The figure depicts the input gradients of models belonging to different regimes of robustness. The numbers above the images indicate the perceptual similarity with the score, as measured by the LPIPS metric. Top row: ImageNet-64x64. Bottom row: CIFAR-10. **Weakly robust** models are accurate but not off-manifold robust. **Bayes-aligned** robust models are accurate and off-manifold robust. These are exactly the models that have perceptually aligned gradients. **Excessively robust** models are excessively on-manifold robust which makes them inaccurate. Best viewed in digital format.

green and blue curves fall in tandem). Remarkably, these results are consistent across the different types of regularization.

## 4.2 Evaluating the Perceptual Alignment of Robust Model Gradients

We now show that the input gradients of an intermediate regime of accurate and relatively off-manifold robust models ("*Bayes-aligned* robust models") are perceptually aligned, whereas the input gradients of *weakly* robust and *excessively* robust models are not. As discussed above, we measure the perceptual similarity of input gradients with the score of the probability distribution. The bottom row of Figure 3 depicts our results on CIFAR-10. For all three robustness objectives, the perceptual similarity of input gradients with the score, as measured by the LPIPS metric, gradually increases with robustness (the orange curves gradually increase). The perceptual similarity then peaks for an intermediate amount of robustness, after which it begins to decrease. Figure 4 depicts our results on ImageNet and ImageNet-64x64. Again, the perceptual similarity of input gradients with the score gradually increases with robustness. On ImageNet-64x64, we also trained excessively robust models that exhibit a decline both in accuracy and perceptual alignment.

To gain intuition for these results, Figure 6, as well as Supplementary Figures 10-15, provide a visualization of model gradients. In particular, Figure 6 confirms that the model gradients belonging to the left and right ends of the curves in Figure 3 are indeed not perceptually aligned, whereas the model gradients around the peak (depicted in the middle columns of Figure 6) are indeed perceptually similar to the score.

While we use the perceptual similarity of input gradients with the score as a useful proxy for the perceptual alignment of input gradients, we note that this approach has a theoretical foundation in the energy-based perspective on discriminative classifiers [33]. In particular, Srinivas and Fleuret [20], Zhu et al. [34] have suggested that the input gradients of softmax-based discriminative classifiers could be related to the score of the probability distribution. To the best of our knowledge, our work is the first to quantitatively compare the input gradients of robust models with independent estimates of the score.

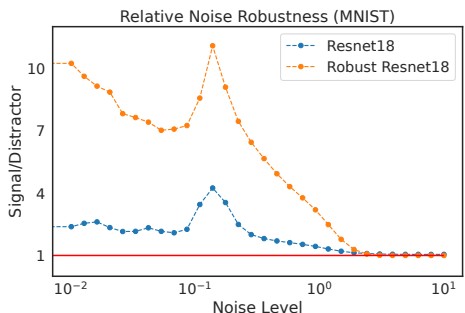

Figure 5: Robust Models are relatively robust to noise on a distractor.

## 4.3 Evaluating Signal *vs.* Distractor Robustness for Robust Models

We now show that robust models are relatively robust to noise on a distractor. Since a distractor is by definition not part of the signal manifold (Section 3.2), this serves as evidence that the input

gradients of robust models are aligned with the signal manifold. In order to control the signal and the distractor, we create a variant of the MNIST dataset where we artificially add the letter "A" as a distractor. This construction is inspired by Shah et al. [11], for details see Supplement Figure 16. Because we know the signal and the distractor by design, we can add noise to only the signal or only the distractor and then measure the robustness of different models towards either type of noise. We call the ratio between these two robustness values the relative noise robustness of the model. Figure 5 depicts the relative noise robustness both for a standard- and an adversarially robust Resnet18. From Figure 5, we see that the standard model is already more robust to noise on the distractor. The robust model, however, is relatively much more robust to noise on the distractor. Since distractor noise is by definition off-manifold robust w.r.t. the signal manifold, this result serves as evidence for Hypothesis 2. Moreover, the input gradients of the robust model (depicted in the supplement) are perceptually aligned. Hence, this experiment also serves as evidence for Hypothesis 1.

## 5  Discussion

**Three Regimes of Robustness.** Our experimental results show that different robust training methods show similar trends in terms of how they achieve robustness. For small levels of robustness regularization, we observe that the classifiers sensitivity to off-manifold perturbations slowly decreases (*weak* robustness), eventually falling below the on-manifold sensitivity, satisfying our key property of *(relative) off-manifold robustness*, as well as alignment with the Bayes classifier (*Bayes-aligned* robustness). Excessive regularization causes models to become insensitive to on-manifold perturbations, which often corresponds to a sharp drop in accuracy (*excessive* robustness).

The observation that robust training consists of different regimes (weak-, Bayes-aligned-, and excessive robustness) calls us to *rethink standard robustness objectives and benchmarks* [35], which do not distinguish on- and off-manifold robustness. An important guiding principle here can be not to exceed the robustness of the Bayes optimal classifier.

**The Limits of Gradient-based Model Explanations.** While Shah et al. [11] find experimentally that gradients of robust models highlight discriminative input regions, we ground this discussion in the signal-distractor decomposition. Indeed, as the gradients of the Bayes optimal classifier are meaningful in that they lie tangent to the signal manifold, we can also expect gradients of robust models to mimic this property. Here the zero gradient values indicates the distractor distribution. However, if a meaningful signal-distractor decomposition does not exist for a dataset, i.e., the data and the signal distribution are identical, then we cannot expect gradients even the gradients of the Bayes optimal classifier to be "interpretable" in the sense that they highlight all the input features as the entire data is the signal.

**Limitations.** First, while our experimental evidence strongly indicates support for the hypothesis that off-manifold robustness emerges from robust training, we do not provide a rigorous theoretical explanation for this. We suspect that it may be possible to make progress on formalizing **Argument 1** in Section 3.3 upon making suitable simplifying assumptions about the model class. Second, we only approximately capture the alignment to the Bayes optimal classifier using the score-gradients from a SOTA diffusion model as a proxy, as we do not have access to a better proxy for a Bayes optimal classifier.

## Acknowledgements

The authors would like to thank Maximilian Müller, Yannic Neuhaus, Usha Bhalla and Ulrike von Luxburg for helpful comments. Sebastian Bordt is supported by the German Research Foundation through the Cluster of Excellence "Machine Learning – New Perspectives for Science" (EXC 2064/1 number 390727645) and the International Max Planck Research School for Intelligent Systems (IMPRS-IS). Suraj Srinivas and Hima Lakkaraju are supported in part by the NSF awards IIS-2008461, IIS-2040989, IIS-2238714, and research awards from Google, JP Morgan, Amazon, Harvard Data Science Initiative, and the Digital, Data, and Design ($D^3$) Institute at Harvard. The views expressed here are those of the authors and do not reflect the official policy or position of the funding agencies.

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

# A  Additional Proofs

**Theorem A.1** (Equivalence between off-manifold robustness and on-manifold alignment). *A function $f : \mathbb{R}^d \to \mathbb{R}$ exhibits on-manifold gradient alignment if and only if it is off-manifold robust wrt normal noise $\mathbf{u} \sim \mathcal{N}(0, \sigma^2)$ for $\sigma \to 0$ (with $\rho_1 = \rho_2$).*

*Proof.* We proceed by observing that we can decompose the input-gradient into on-manifold and off-manifold components by projecting onto the tangent space and its orthogonal component respectively, i.e., $\nabla_{\mathbf{x}} f(\mathbf{x}) = \mathbb{P}_x \nabla_{\mathbf{x}} f(\mathbf{x}) + \mathbb{P}_x^{\perp} \nabla_{\mathbf{x}} f(\mathbf{x})$.

We also observe that we can write the gradient norm in terms of an expected dot product, i.e., $\frac{1}{\sigma^2} \mathbb{E}_{\mathbf{u} \sim \mathcal{N}(0,\sigma^2)} (\nabla_{\mathbf{x}} f(\mathbf{x})^{\top} \mathbf{u})^2 = \frac{1}{\sigma^2} \nabla_{\mathbf{x}} f(\mathbf{x})^{\top} \mathbb{E}(\mathbf{u}\mathbf{u}^{\top}) \nabla_{\mathbf{x}} f(\mathbf{x}) = \|\nabla_{\mathbf{x}} f(\mathbf{x})\|^2$.

Using these facts we can compute the norm of the off-manifold component as follows,

$$\underbrace{\frac{\|\nabla_{\mathbf{x}} f(\mathbf{x}) - \mathbb{P}_x \nabla_{\mathbf{x}} f(\mathbf{x})\|^2}{\|\nabla_{\mathbf{x}} f(\mathbf{x})\|^2}}_{\text{On-manifold gradient alignment}} = \frac{\|\mathbb{P}_x^{\perp} \nabla_{\mathbf{x}} f(\mathbf{x})\|^2}{\|\nabla_{\mathbf{x}} f(\mathbf{x})\|^2}$$

$$= \frac{\frac{1}{\sigma^2} \mathbb{E}_{\mathbf{u}_{\text{off}} \sim \mathcal{N}(0,\sigma^2 \Sigma)} (\nabla_{\mathbf{x}} f(\mathbf{x})^{\top} \mathbf{u}_{\text{off}})^2}{\frac{1}{\sigma^2} \mathbb{E}_{\mathbf{u} \sim \mathcal{N}(0,\sigma^2)} (\nabla_{\mathbf{x}} f(\mathbf{x})^{\top} \mathbf{u})^2} \quad ; \quad \Sigma = \text{Cov}(\mathbf{u}_{\text{off}}) = \mathbb{P}_x^{\perp} (\mathbb{P}_x^{\perp})^{\top}$$

$$= \lim_{\sigma \to 0} \underbrace{\frac{\mathbb{E}_{\mathbf{u}_{\text{off}} \sim \mathcal{N}(0,\sigma^2 \Sigma)} (f(\mathbf{x} + \mathbf{u}_{\text{off}}) - f(\mathbf{x}))^2}{\mathbb{E}_{\mathbf{u} \sim \mathcal{N}(0,\sigma^2)} (f(\mathbf{x} + \mathbf{u}) - f(\mathbf{x}))^2}}_{\text{Off-manifold robustness}}$$

The second line is obtained by using the fact above regarding re-writing the gradient norm in terms of the expected dot product, and the final line is obtained by using a first order Taylor expansion, which is exact in the limit of small sigma. From the equality of first and last terms, we have that the on-manifold gradient alignment $\Leftrightarrow$ the off-manifold robustness. $\qquad \square$

**Theorem A.2.** *The input-gradients of Bayes optimal classifiers lie on the signal manifold $\Leftrightarrow$ Bayes optimal classifiers are relative off-manifold robust.*

*Proof.* From definition 3, it is clear that given a classification problem, there exists a single distractor distribution $d(\mathbf{x})$. Now, we take gradients of log probabilities of the Bayes optimal classifiers, which results in:

$$\nabla_{\mathbf{x}} \log p(y = i \mid \mathbf{x}) = \nabla_{\mathbf{x}} \log p(\mathbf{x} \mid y = i) - \sum_j p(y = j \mid \mathbf{x}) \nabla_{\mathbf{x}} \log p(\mathbf{x} \mid y = j)$$

We notice first that the vectors $\nabla_{\mathbf{x}} \log p(\mathbf{x} \mid y)$ all lie tangent to the data manifold by definition, as this data generating process $p(\mathbf{x} \mid y)$ itself defines the data manifold. As $\nabla_{\mathbf{x}} \log p(y \mid \mathbf{x})$ is a *linear combination* of the class-conditional generative model gradients, it follows that the input-gradient of the Bayes optimal model also lie tangent to the data manifold. Now, like any vector on the tangent space at $\mathbf{x}$, it can be decomposed into signal and distractor components. Computing the distractor, we find that

$$\nabla_{\mathbf{x}} \log p(y \mid \mathbf{x}) \odot (1 - \mathbf{m}^*(\mathbf{x})) = d(\mathbf{x}) - \sum_j p(y = j \mid \mathbf{x}) d(\mathbf{x}) = 0$$

This happens because the distractor is independent of the label, thus the distractor component is zero, and the input-gradient of the Bayes optimal model lies entirely on the signal manifold. From Theorem A.1, it follows that when a model gradients lie on a manifold, it is also off-manifold robust wrt that manifold.

$\qquad \square$

# B Experimental Details

## B.1 Robust Training Objectives

We consider the following robust training objectives, where $l(x, y)$ denotes the cross-entropy loss function.

1. Gradient norm regularization: $l(f(x), y) + \lambda\|\nabla_{\mathbf{x}} f(\mathbf{x})\|_2^2$ with a regularization constant $\lambda$.

2. A smoothness penalty: $l(f(x), y) + \lambda\mathbb{E}_{\epsilon\sim\mathcal{N}(0,\sigma^2)}\|f(x + \epsilon) - f(x)\|_2^2$ with a fixed noise level $\sigma^2$ and a varying regularization constant $\lambda$.

3. Randomized Smoothing: $\mathbb{E}_{\epsilon\sim\mathcal{N}(0,\sigma^2)}l(f(x + \epsilon), y)$ with a noise level $\sigma^2$.

4. Adversarial Robust Training: $l(f(\tilde{x}), y)$ where $\tilde{x} = \arg\max_{\tilde{x}\in B_\epsilon(x)} l(f(\tilde{x}), y)$ and $\tilde{x}$ was obtained from the $\epsilon$-ball around $x$ using projected gradient descent.

## B.2 Training Details

On CIFAR-10, we trained Resnet18 models for 200 epochs with an initial learning rate of 0.025. When training with gradient norm regularization or the smoothness penalty and large regularization constants we reduced the learning rate proportional to the increase in the regularization constant. After 150 and 175 epochs, we decayed the learning rate by a factor of 10.

On ImageNet-64x64, we trained Resnet18 models for 90 epochs with a batch size of 4096 and an initial learning rate of 0.1 that was decayed after 30 and 60 epochs, respectively. We used the same parameters for projected gradient descent (PGD) as in [29], that is we took 3 steps with a step size of $2\epsilon/3$.

On the MNIST dataset with a distractor, we trained a Resnet18 model for 9 epochs with an initial learning rate of 0.1 that was decayed after 3 and 6 epochs, respectively. We also trained an $l_2$-adversarially robust Resenet18 with projected gradient descent (PGD). We randomly chose the perturbation budget $\epsilon \in \{1, 4, 8\}$ and took 10 steps with a step size of $\alpha = 2.5\epsilon/10$.

## B.3 Diffusion Models

On CIFAR-10, we use the unconditional diffusion model `edm-cifar10-32x32-uncond-vp`. For comparison, Figure Supplement Figure 17 presents the results with the conditional diffusion model `edm-cifar10-32x32-cond-vp`. On ImageNet-64x64, we use the conditional diffusion model `edm-imagenet-64x64-cond-adm`. All models are available at `https://github.com/NVlabs/edm`. The noise levels $\sigma = 0.5$ and $\sigma = 1.2$ were chosen to maximize the perceptual quality of the estimate of the score obtained from the diffusion model before running the experiments with the LPIPS metric.

## B.4 Conditional- and Unconditional Diffusion Models

Based on our theory, there are two principled ways to evaluate diffusion models and discriminative classifiers. The first is to take the input gradient with respect to a single class and compare it to a conditional diffusion model. The second is to sum the input gradients of all the classes and compare it to an unconditional diffusion model (i.e., computing marginal probabilities). The main paper present the first approach on ImageNet and ImageNet-64x-64 and the second approach on CIFAR-10. Supplement Figure 17 presents the first approach on CIFAR-10. We observe that there is no qualitative difference, i.e. both approaches give the same result.

## B.5 Model Gradients

With the unconditional diffusion model, we sum the input gradients across all classes. With the conditional diffusion model, we consider the input gradient with respect to the predicted class. We consider input gradients before the softmax [20].

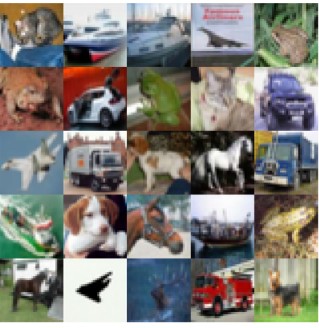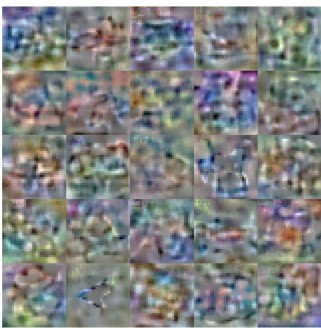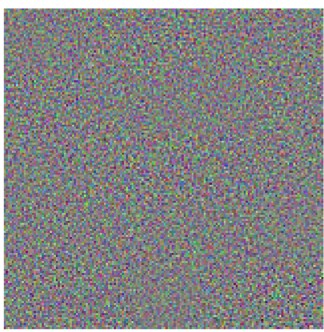

Figure 7: **Left:** Images from CIFAR10. **Middle:** Random perturbations on the data manifold. **Right:** Random perturbations off the data manifold.

### B.6 CIFAR-10 Autoencoder

We use `https://github.com/clementchadebec/benchmark_VAE` to train an autoeoncoder on CIFAR-10 with a latent dimension $k = 128$. We use a default architecture and training schedule. We then use the autoencoder to estimate, at each data point, a 128-dimensional tangent space. Figure 7 depicts random directions within the estimated tangent spaces.

### B.7 Pre-Trained Robust Models on ImageNet

On ImageNet, we use the pre-trained robust Resnet18 models form `https://github.com/microsoft/robust-models-transfer`. To load these models, we use the robustness library `https://github.com/MadryLab/robustness`.

### B.8 Estimating the Score on ImageNet

We estimate the score on ImageNet using the diffusion model for ImageNet-64x64. To estimate the score, we simply down-scale an image to 64x64.

### B.9 MNIST with a Distractor

The MNIST data set with a distractor is inspired by [11]. The data set consists of gray-scale images of size 56x28. Every image contains a single MNIST digit and the distractor. We choose the fixed letter "A" as the distractor. On every image, we randomly place the distractor on top or below the MNIST digit. In order to estimate the relative noise robustness, we separately add different levels of noise to the signal or distractor. Figure 16 depicts images and models gradients on this data set.

### B.10 The LPIPS metric

The LPIPS metric measures the perceptual similarity between two different images. The metric itself corresponds to a loss, meaning that lower values correspond to more similar images [31]. The figures in the main paper depict 1-LPIPS, that is higher values correspond to more similar images.

## C Additional Plots

The figures below depict the model gradients of different types of models, ranging from weakly robust to excessively robust. The figures depict the relationship between model gradients and the score qualitatively. This complements the quantitative results in the main paper.

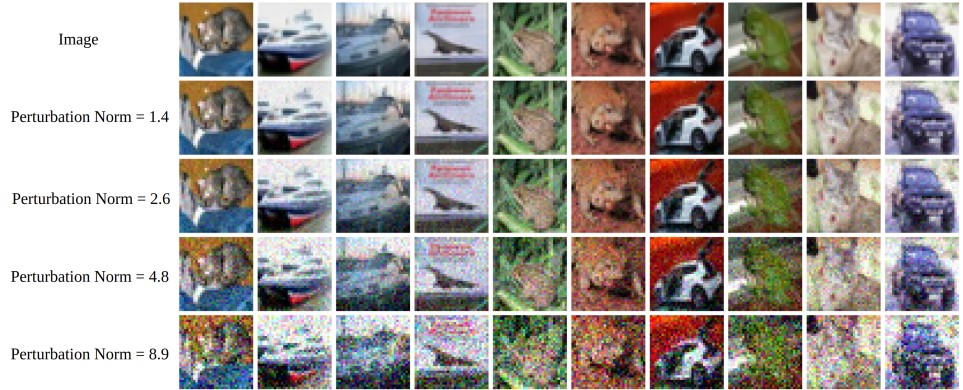

Figure 8: Perturbations of different size, as measured by the $L_2$-norm of the perturbation. The Figure depicts the original images with random perturbations of the given $L_2$-norm.

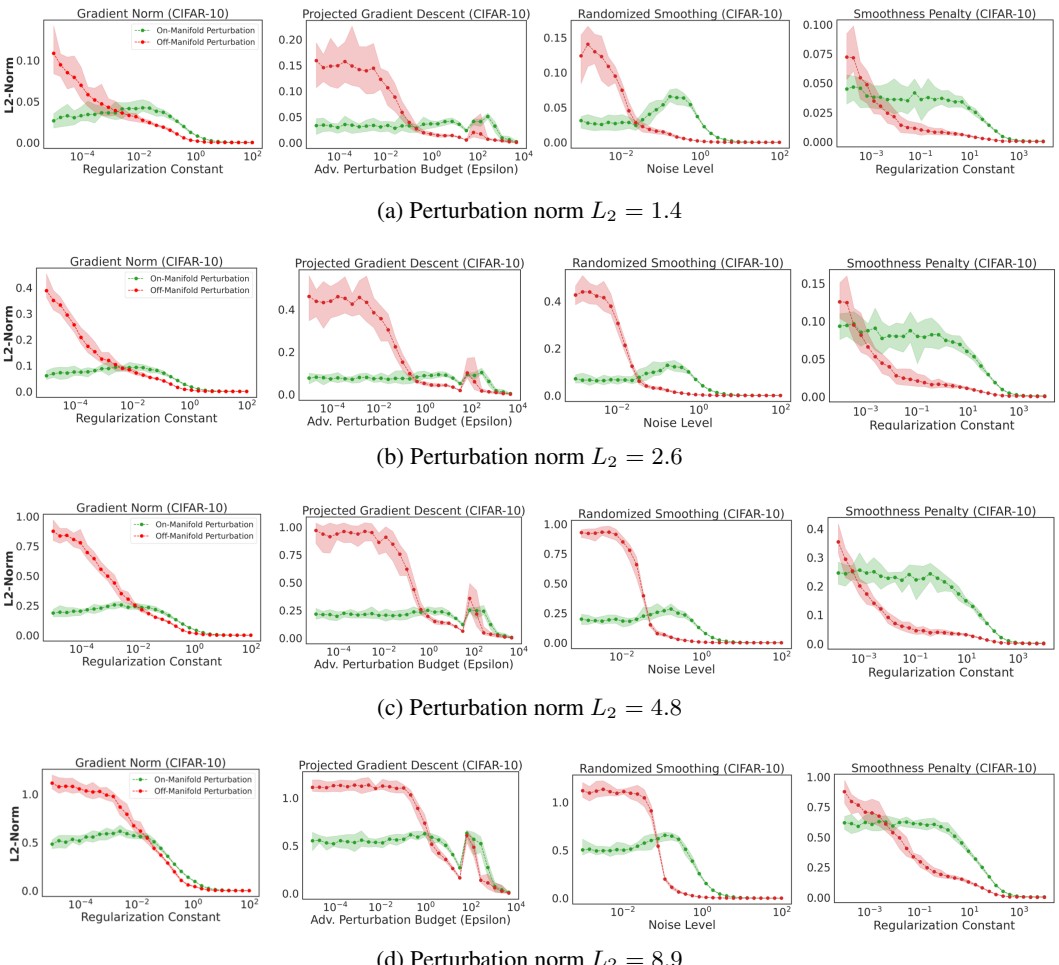

Figure 9: On- and off-manifold robustness of Resnet18 models trained with different objectives on CIFAR-10 (larger values correspond to less robustness). The Figure depicts the on- and off-manifold robustness for perturbations of different size, as measured by the $L_2$-norm of the pertubation. Compare Figure 3 in the main paper.

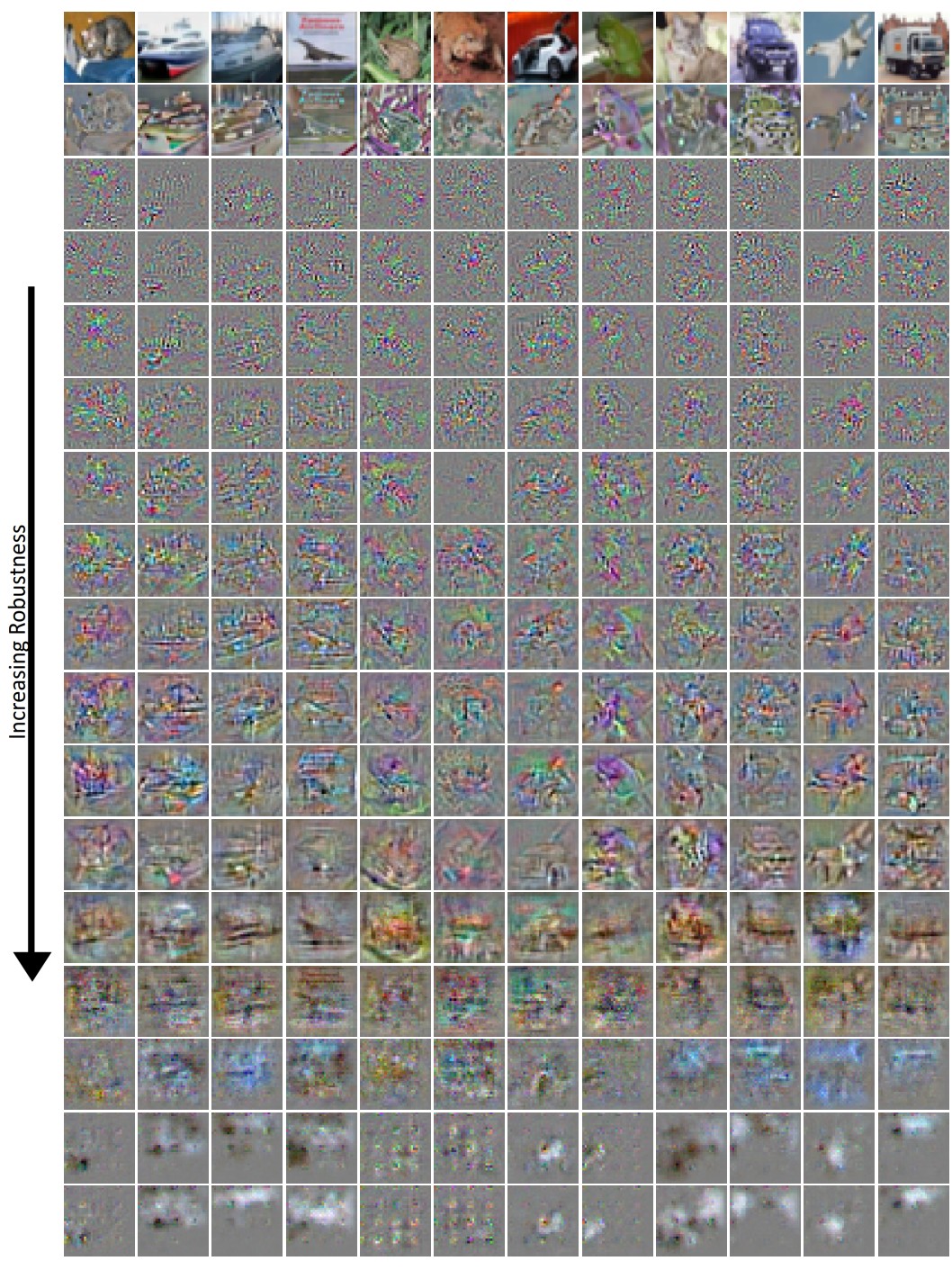

Figure 10: The input gradients of different models trained with **gradient norm regularization on CIFAR-10**. The top rows depict the image, the score, and the input gradients of unrobust models. The middle rows depict the perceptually aligned input gradients of robust models. The bottom rows depict the input gradients of excessively robust models. Best viewed in digital format.

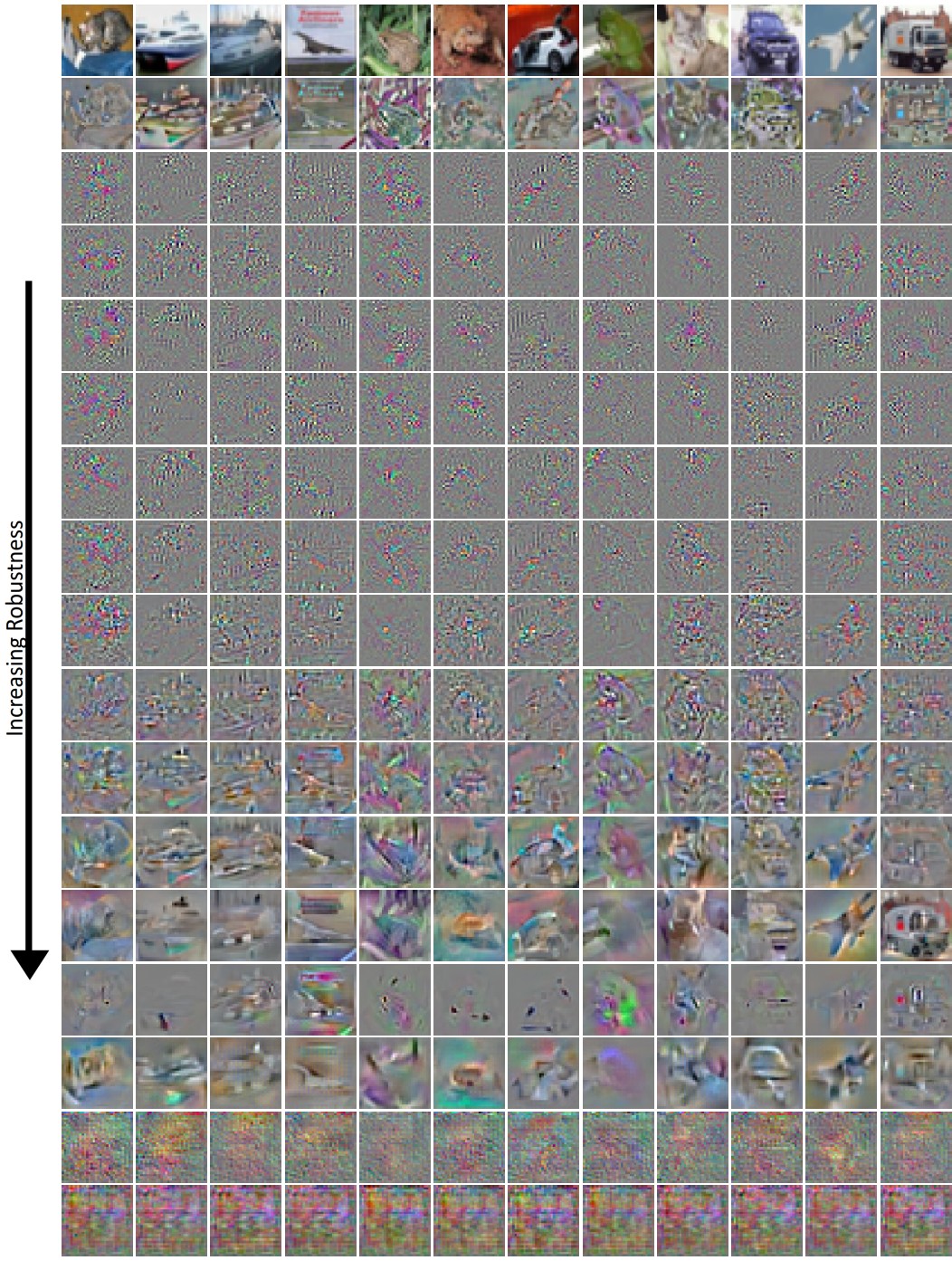

Figure 11: The input gradients of different models trained with **projected gradient descent on CIFAR-10**. The top rows depict the image, the score, and the input gradients of unrobust models. The lower middle rows depict the perceptually aligned input gradients of robust models. The bottom rows depict the input gradients of excessively robust models. Best viewed in digital format.

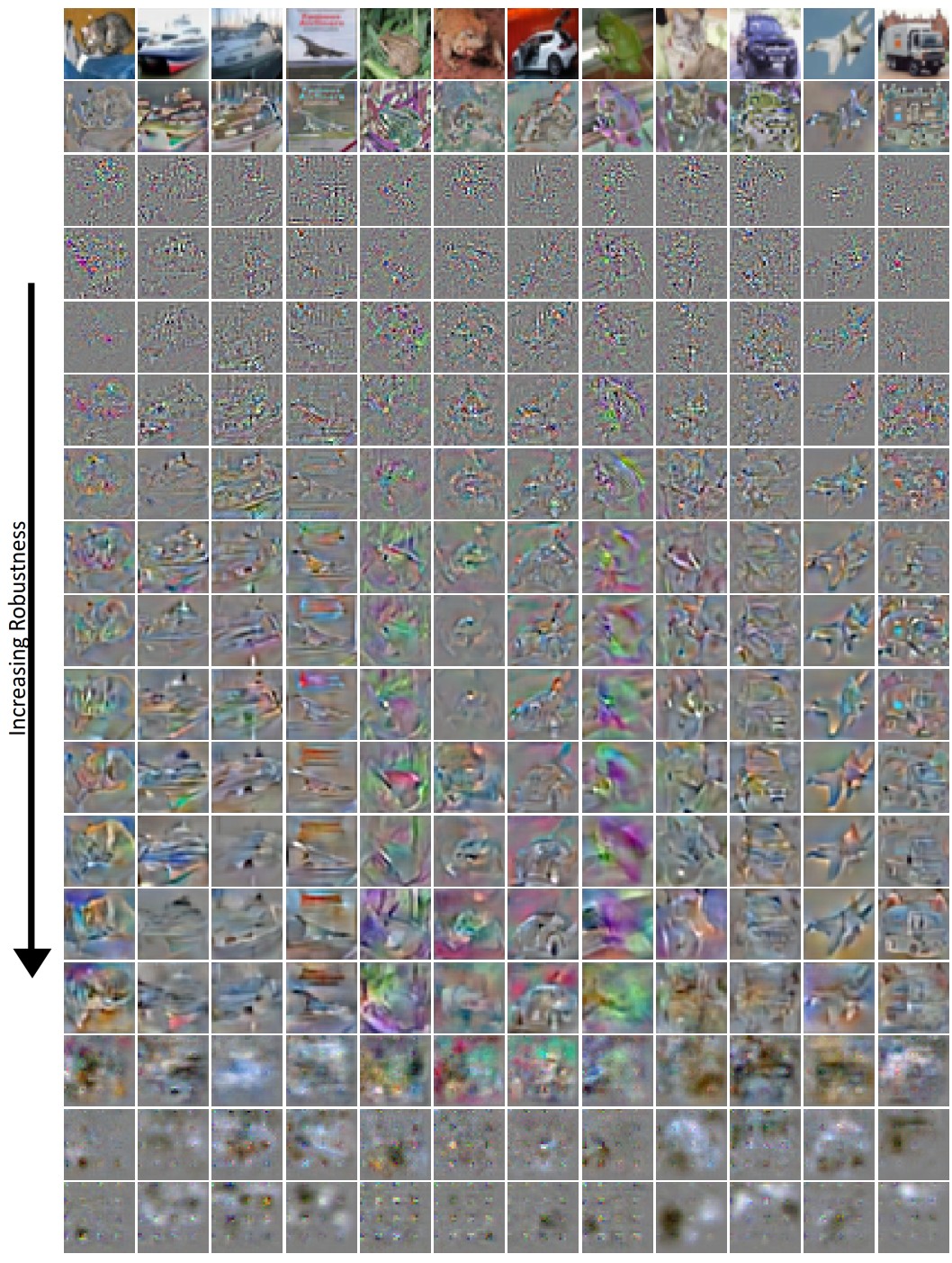

Figure 12: The input gradients of different models trained with a **smoothness penalty on CIFAR-10**. The top rows depict the image, the score, and the input gradients of unrobust models. The middle rows depict the perceptually aligned input gradients of robust models. The bottom rows depict the input gradients of excessively robust models. Best viewed in digital format.

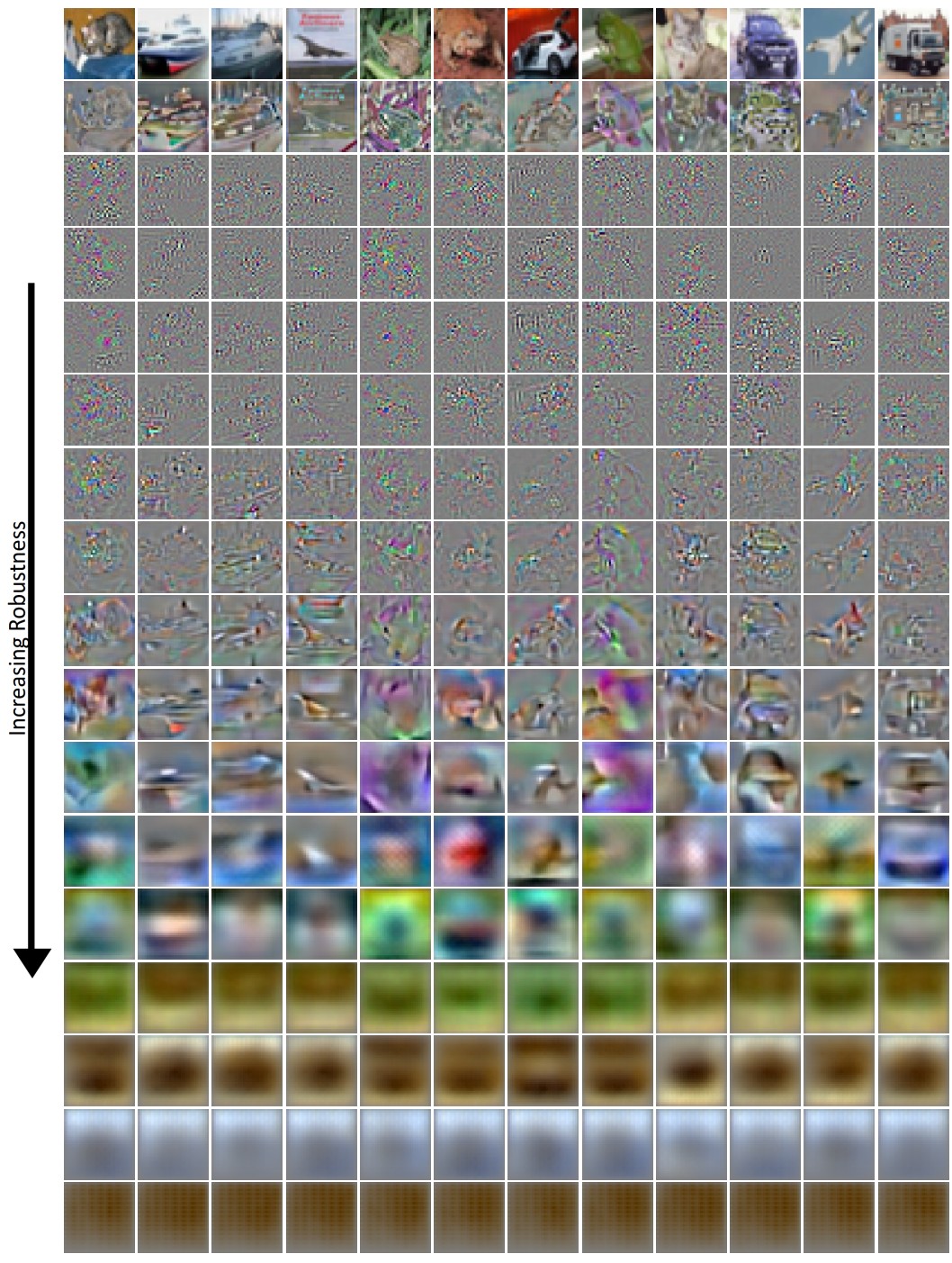

Figure 13: The input gradients of different models trained with **randomized smoothing on CIFAR-10**. The top rows depict the image, the score, and the input gradients of unrobust models. The middle rows depict the perceptually aligned input gradients of robust models. The bottom rows depict the input gradients of excessively robust models. Best viewed in digital format.

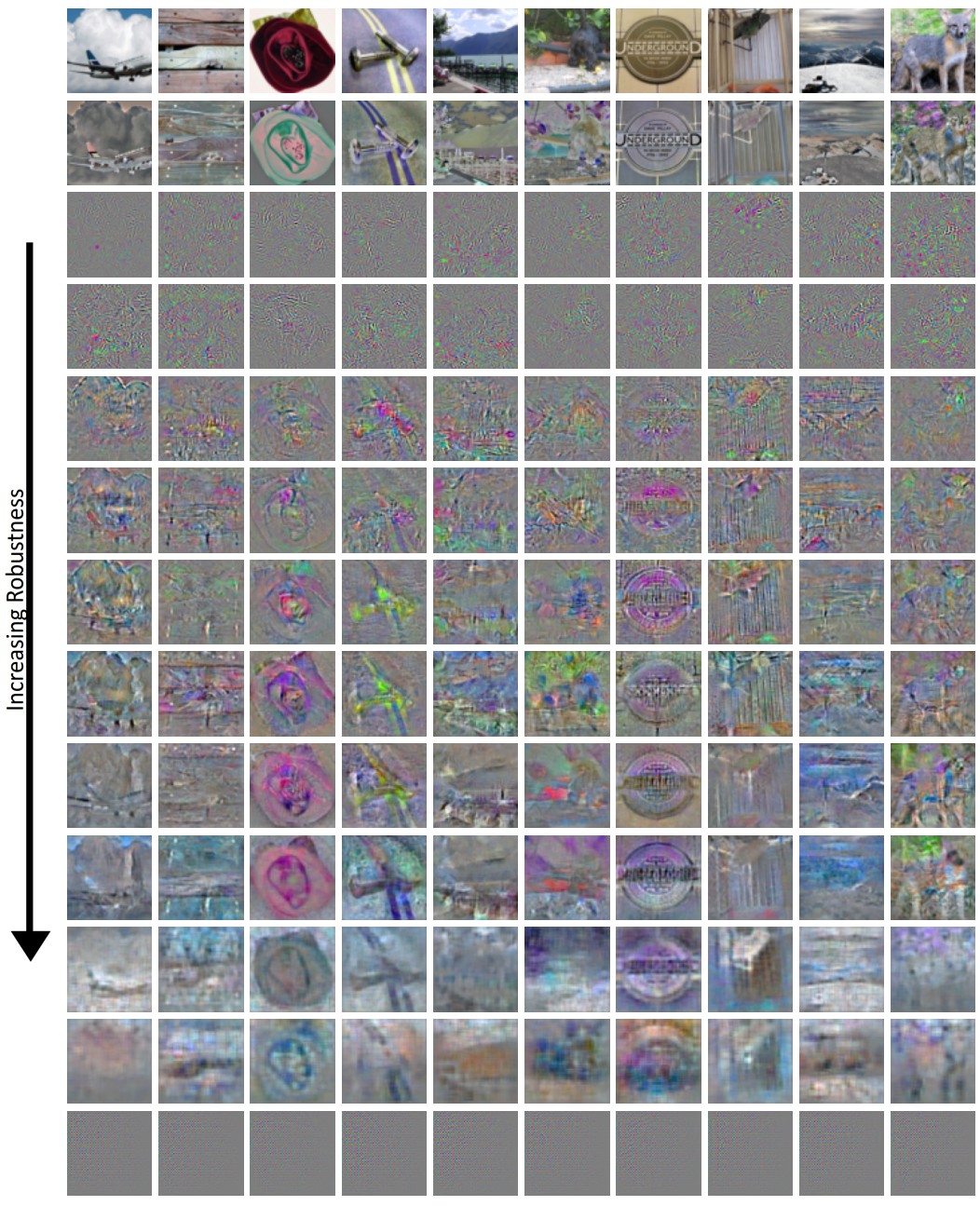

Figure 14: The input gradients of different models trained with **projected gradient descent on ImageNet-64x64**. The top rows depict the image, the score, and the input gradients of unrobust models. The middle rows depict the perceptually aligned input gradients of robust models. The bottom rows depict the input gradients of excessively robust models. Best viewed in digital format.

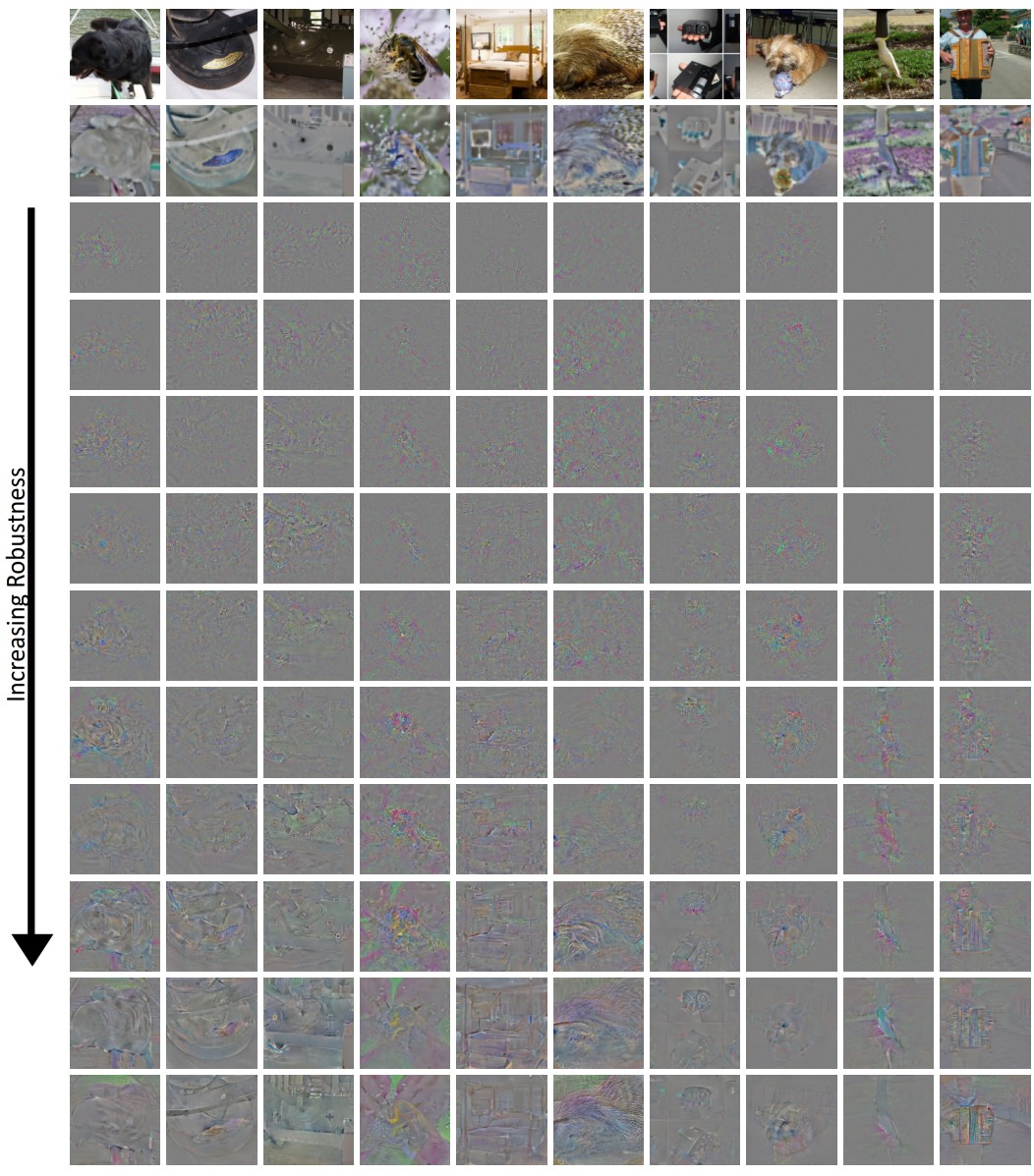

Figure 15: The input gradients of different models trained with **projected gradient descent on ImageNet**. The models are from [29]. The top rows depict the image, the score, and the input gradients of unrobust models. The bottom rows depict the perceptually aligned input gradients of robust models. Best viewed in digital format.

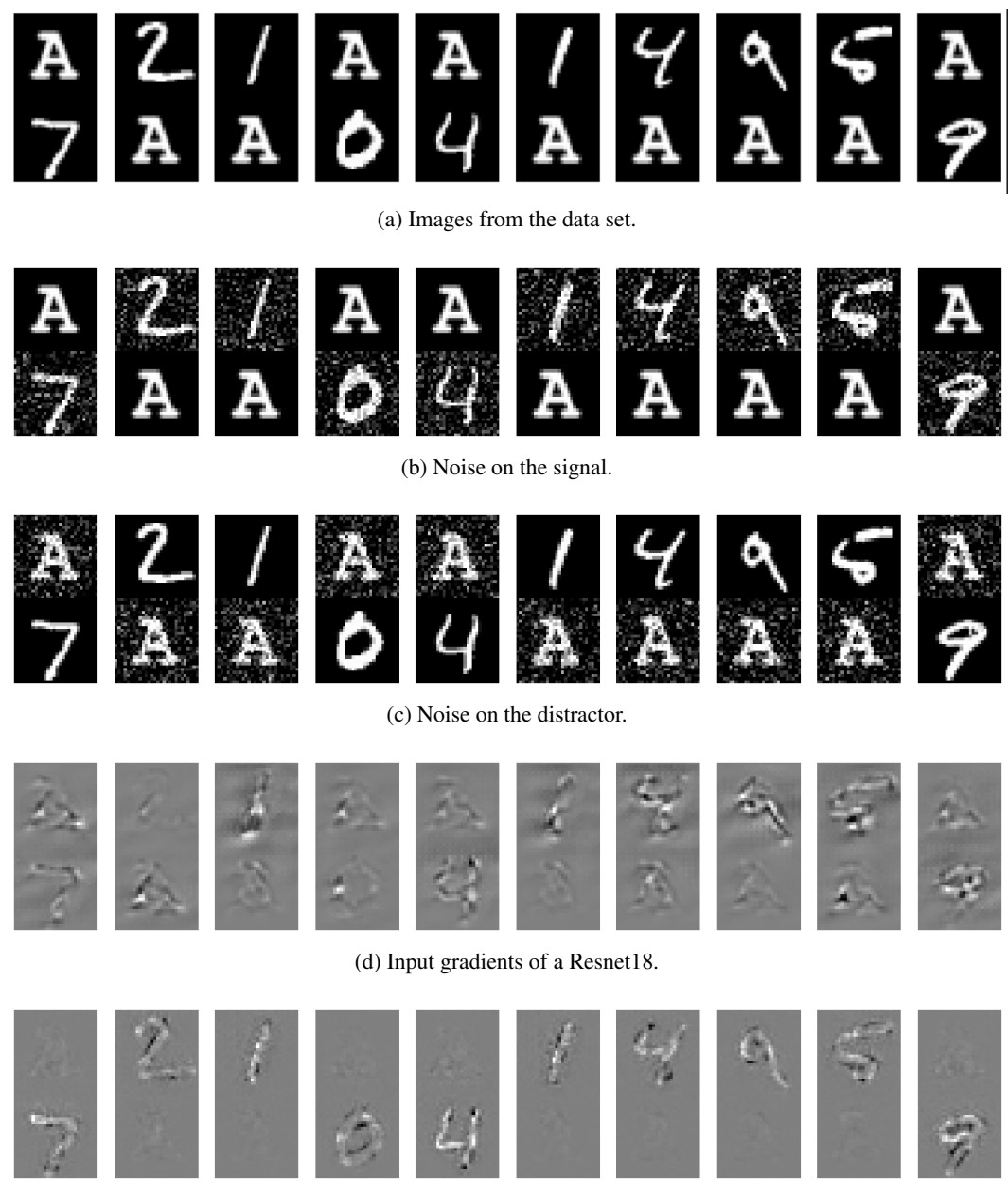

(a) Images from the data set.

(b) Noise on the signal.

(c) Noise on the distractor.

(d) Input gradients of a Resnet18.

(e) Input gradients of an adversarially robust Resnet18.

Figure 16: The MNIST dataset with a distractor used to create Figure 5 in the main paper.

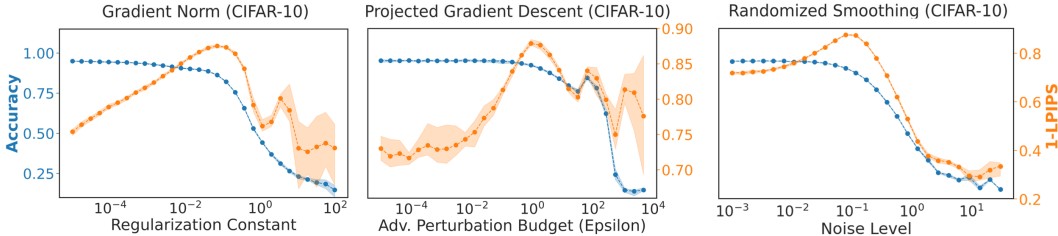

Figure 17: Measuring the perceptual alignment between the input gradients of Resnet18 models and the score as estimated with a **conditional** diffusion model. Compare Supplement Section B.4 and Figure 3 in the main paper. The Figure in the main paper depicts results with an **unconditional** diffusion model.

