# OpenReview forum: "Which Models have Perceptually-Aligned Gradients? An Explanation via Off-Manifold Robustness"
_NeurIPS.cc/2023/Conference — NeurIPS 2023 spotlight_

### Official Review · Reviewer_eaFT · 2023-06-26

**Soundness:** 4 excellent
**Presentation:** 4 excellent
**Contribution:** 4 excellent
**Rating:** 8
**Confidence:** 5

**Summary:**

This paper presents a theoretical characterization and some theorems supporting previous empirical finidings on perceptually aligned gradients (PAG) of classification neural network models. Specifically, it provides the first rigorous definitions for the previously qualitative definitions of "PAG", and provides the first attempt at a theoretical framework connecting PAG and model robustness, which has been observed empricially in previous work.

**Strengths:**

- Very well-written and well-presented paper. Covers and taxonomizes the relevant literature very nicely.
- Math and definitions are intuitive, relevant, and sound. I especially like the fact that the theorem assumes almost nothing on the model behavior or training objective.
- Overall, this paper addresses a very important and relevant topic, characterizes a lot of themes in it, and provides a great theoretical explanation for it. This theoretical justification was notably missing from previous PAG literature.
- Empirical evaluation metrics are carefully picked and well-thought-out. Comparing PAG to score-based models is especially relevant (this was first suggested in [6], I think a reference is due on lines 264-274).

**Weaknesses:**

- lines 106-107: noise can also have both on- and off- manifold elements (think of a linear combination of two on- and off- manifold noise vectors). Therefore, the "otherwise" on line 107 is incorrect.
- Minor issue: Defnition 1 is missing a definition of x (for formality).
- While the math (theorem 1) is nice, it does not consider adversarially chosen noise. It would be nice to mention that for normal noise with $\sigma \rightarrow 0$, this covers adversarial noise as well, but it should be made clear that more work is needed to specifically consider adversarial attacks.
- Line 151: Paragraph title mentions difference between data and signal manifolds, but the paragraph does not discuss "data manifold". Typo?
- The definition of signal vs distractor is too simplistic. Signals and distractors are not necessarily separated by a pixel mask only. The underlying signal manifold could lie on a linear or non-linear projection of x, not necessarily on the pixel space itself. This weakens the resulting math (theorem 2), unless the authors can rewrite their proofs w.r.t. a determinstic "signal" function that is generalized beyond simple point-wise masking.

**Questions:**

- In the abstract, is noise augmentation and randomized smoothing the same thing? If so, I would suggest listing them as one item and not two.
- Lines 272-274: Is the score-based model used here class-conditional? If so, it should be made clear by defining p(x|y) and not p(x) in the score function definitions. If not, further explanation is needed. Why would we characterize perceptual alignment using a class-unaware model? "PAG" usually refers to salient class-specific features.

**Limitations:**

Yes

---

> ### Author Rebuttal · Authors · 2023-08-09
>
> Thanks for your insightful review! We're glad you liked the paper and found the theory intuitive and relevant. We address your questions below.
>
> 1.*“While the math (theorem 1) is nice, it does not consider adversarially chosen noise. It would be nice to mention that for normal noise with $\sigma \rightarrow 0$, this covers adversarial noise as well, but it should be made clear that more work is needed to specifically consider adversarial attacks.”*
>
> Good point! We think adversarial noise are definitely an important case, but slightly tricky to handle as it represents the worst-case perturbation and has measure zero. For example, it can still happen that models may be less robust off-manifold compared to on-manifold, but the only adversarial noise is on-manifold. Essentially, this can happen because adversarial noise is about misclassification, but our notions of robustness are about change in output value. But, note that this theory nonetheless holds for adversarially-trained models in practice, which have on-manifold gradients and hence off-manifold robustness in the average case. Please let us know in case you have more thoughts about this problem!
>
> ---
>
> 2.*“The definition of signal vs distractor is too simplistic. Signals and distractors are not necessarily separated by a pixel mask only. The underlying signal manifold could lie on a linear or non-linear projection of x, not necessarily on the pixel space itself. This weakens the resulting math (theorem 2), unless the authors can rewrite their proofs w.r.t. a determinstic "signal" function that is generalized beyond simple point-wise masking.”*
>
> We agree that Definition 3 can be extended such that the signal and distractor are any pointwise orthogonal complements of each other, and need not be axis aligned to the input pixels. The reason we define these in this manner is to explain the observations of Shah et al. (“Do input gradients highlight discriminative features?”, NeurIPS 2021) who observe that input gradients of robust models highlight discriminative features, which they define as input pixels that can be used to predict the label, which serves as the basis for "phenomenon 1" in our paper. Having said that, we agree that generalizing definition 3 makes the theory potentially more interesting and general. We will add a note about this in the paper.
>
> ---
>
> 3.*“In the abstract, is noise augmentation and randomized smoothing the same thing? If so, I would suggest listing them as one item and not two.”*
>
> By “noise augmentation” we mean the smoothness penalty detailed in Section C.1 in the Supplement, which is different from randomized smoothing. Essentially, this involves training models with an explicit regularization encouraging the model behaviour to be similar with and without noise, i.e., regularizing $ || f(x)  - f(x + \epsilon) ||^2$ to be small, where $\epsilon \sim \mathcal{N}(0, \sigma^2)$.
>
> ---
>
> 4.*“Lines 272-274: Is the score-based model used here class-conditional? If so, it should be made clear by defining p(x|y) and not p(x) in the score function definitions. If not, further explanation is needed. Why would we characterize perceptual alignment using a class-unaware model? "PAG" usually refers to salient class-specific features.”*
>
> The diffusion model is unconditional. Interestingly, we found little difference between the scores of class-conditional and unconditional diffusion models on natural images (comparing “cifar10-32x32-uncond-vp” and “cifar10‑32x32‑cond‑vp” from [10]). Empirically, our results are robust towards using conditional and unconditional diffusion models.
>
> Thank you also for all the suggestions with the typos, we will make these corrections in the draft.

---

> > ### Comment · Reviewer_eaFT · 2023-08-12
> >
> > Thank you for the response. I maintain my score of 8.

---

### Official Review · Reviewer_aEc6 · 2023-07-01

**Soundness:** 4 excellent
**Presentation:** 4 excellent
**Contribution:** 3 good
**Rating:** 7
**Confidence:** 5

**Summary:**

This paper studies Perceptually Aligned Gradients (PAGs), a phenomenon where the input gradients are semantically meaningful and aligned with human perception.
While this trait gained research attention, we do not truly understand it.
To this end, the paper proposes an explanation via off-manifold robustness.
Namely, they show empirically and theoretically that models that are off-manifold robust have perceptually aligned gradients for different robustification techniques.
Moreover, they propose a quantitative way to assess PAG, while prior to this work, PAG was evaluated qualitatively only.
In addition, they identify three levels of robustness and analyze them in terms of PAG.

**Strengths:**

* Perceptually Aligned Gradients is a fascinating phenomenon, and despite gaining substantial research attention, we do not understand it. Shedding light on this trait is an interesting and important research goal. The paper empirically and theoretically verifies the relationship between PAG and off-manifold robustness. To strengthen their findings, they do so for different robustification techniques.

* Decomposing the gradients of a model to on and off-manifold directions is very interesting thinking. Besides being theoretically logical, they also develop a way of doing so.

* Excellent and clear writing. The authors have done a good job of providing motivation for this work. In addition, while several papers use this term to describe different behaviors, this paper formulates the definitions and makes an order in the various definitions.


**Weaknesses:**

* The connection between a conditional score function and PAG was made in [1]. Despite not being used for assessing PAG, they show that a classifier trained to mimic the score function possesses PAG. Thus, I do not find linking the two for assessing PAG very novel.

* Overstatements and bit trivial claims - “In this work, we provide 7 a first explanation of PAGs” [abstract], but [4] offers an explanation of the generative capabilities of adversarial training using energy-based models. Additionally, I find some claims that the authors made efforts to explain quite straightforward. For example, robust models are off-manifold robust. We know robust models have PAG, meaning their gradients are aligned on the manifold. This means that the change of loss in off manifold direction is smaller than the on-manifold (because otherwise, the gradients weren’t aligned), making such models off-manifold robust.

**Minor weaknesses and requests**

* I find this important to report the LPIPS results in Figure 5 to demonstrate that the used metric indeed makes sense and aligns with the human evaluation of PAG.

* Missing citations - I think that in order to motivate the significance of studying PAG for the general audience, it is important to include more works that study and utilize PAG. For instance, [2,3].

[1] Do Perceptually Aligned Gradients Imply Robustness?

[2] On the benefits of models with perceptually-aligned gradients.

[3] BIGRoC: Boosting Image Generation via a Robust Classifier.

[4] Towards Understanding the Generative Capability of Adversarially Robust Classifiers.








**Questions:**

* Is the score function being modeled in the measuring Perceptual Alignment paragraph for clean images or noisy images? From my understanding, diffusion networks model the noisy score function and not the clean one.

* What is the rationale for measuring the LPIPS for gradients? It is usually used for images, and input-gradient distribution is significantly different. Why does it make sense to use this rather than cosine similarity or other closed-form metrics?

* Figure 2 shows that there are points where models are good in off-manifold robustness before being on-manifold robust (for example, the left figure in the top row of Figure 2, around a noise level of 0.5). Can this analysis help us find robust models that have better robustness-accuracy tradeoffs?

* Why are the results of Imagenet 64 so low?

In general, I find this paper interesting and would like to increase my score given the authors' response.


**Limitations:**

The authors have adequately addressed the limitations.

---

> ### Author Rebuttal · Authors · 2023-08-09
>
> Thanks for your great review, and we're glad you found our paper interesting! We address specific comments below.
>
> 1.*“The connection between a conditional score function and PAG was made in [1]. Despite not being used for assessing PAG, they show that a classifier trained to mimic the score function possesses PAG. Thus, I do not find linking the two for assessing PAG very novel.”*
>
> The reviewer is right that score-functions have been used in the context of PAGs before [1], but to the best of our knowledge, we are the first to use score-functions to provide a quantitative metric to evaluate PAGs.
>
> ---
>
> 2.*“Overstatements and bit trivial claims - “In this work, we provide a first explanation of PAGs” [abstract], but [4] offers an explanation of the generative capabilities of adversarial training using energy-based models."*
>
> Thanks for the reference! While we believe that the provided reference [4] is indeed a valuable contribution, it misses a critical component of the PAGs phenomenon, that gradients of robust models highlight only the discriminative components (see Phenomenon 1, line 38 in our paper) while ignoring the rest. However we agree that it can be a partial explanation for Phenomenon 2, i.e., generative capabilities, and hence we shall re-word our claims accordingly.
>
> ---
>
> 3.*"Additionally, I find some claims that the authors made efforts to explain quite straightforward. For example, robust models are off-manifold robust. We know robust models have PAG, meaning their gradients are aligned on the manifold. This means that the change of loss in off manifold direction is smaller than the on-manifold (because otherwise, the gradients weren’t aligned), making such models off-manifold robust.”*
>
> To the best of our knowledge, it has not been well-established in prior works that PAGs imply gradients lie on-manifold (Hypothesis 2), and our paper takes a step in this direction. While theorem 1 is intuitive, it is critical to explaining PAGs as it helps link PAGs to model robustness, which is our main objective. In addition, theorem 1 only applies in very limited settings (under the limit of small Gaussian noise), and the contribution of our work is to show that this also holds for realistic noise levels and for real models.
>
> ---
>
> 4.*“I find this important to report the LPIPS results in Figure 5 to demonstrate that the used metric indeed makes sense and aligns with the human evaluation of PAG.”*
>
> Great point. Please see the pdf file provided as part of the rebuttal. The reviewer can also compare Figure 2 with Supplement Figures 7-11.
>
> ---
>
> 5.*“Missing citations - I think that in order to motivate the significance of studying PAG for the general audience, it is important to include more works that study and utilize PAG. For instance, [2,3].”*
>
> [2] is already cited in the paper, thanks for pointers to [3,4]. We shall discuss applications of PAGs in the paper.
>
> ---
>
> 6.*"Is the score function being modeled in the measuring Perceptual Alignment paragraph for clean images or noisy images? From my understanding, diffusion networks model the noisy score function and not the clean one.”*
>
> Diffusion models provide a family of score functions parametrized by the noise level sigma. In theory, noise training in these models ensures scores are well-defined everywhere. Once trained, the score-function is applicable to all points in the input with an appropriate sigma. We choose the parameter sigma to maximize the perceptual alignment of the score.
>
> ---
>
> 7.*“What is the rationale for measuring the LPIPS for gradients? It is usually used for images, and input-gradient distribution is significantly different. Why does it make sense to use this rather than cosine similarity or other closed-form metrics?”*
>
> We choose the LPIPS metric because we find that it adequately quantifies the observed variation in the perceptual similarity between the score and the input gradients of differently robust models. Independently of the chosen metric, this variation is observed in the qualitative depictions in Supplement Figures 7-11. We agree with the reviewer that the LPIPS metric is not necessarily perfect and that other metrics could be considered. Empirically, however, we found that the LPIPS metric worked best. For example, we did initial experiments with the cosine similarity and found the results, while often qualitatively similar, to be less robust and less aligned with our human perception. A potential reason for this might be that the cosine similarity is rather sensitive to the fact that the score aligns with the data manifold, whereas model gradients align with the signal manifold.
>
> ---
>
> 8.*“Figure 2 shows that there are points where models are good in off-manifold robustness before being on-manifold robust (for example, the left figure in the top row of Figure 2, around a noise level of 0.5). Can this analysis help us find robust models that have better robustness-accuracy tradeoffs?”*
>
> Great question! We would argue that an analysis of the on- and off-manifold robustness curves can help us identify models that are relatively accurate and also have perceptually-aligned gradients (or off-manifold robustness). Considering the example pointed out by the reviewer (noise level of 0.5), this corresponds to a model with relatively high accuracy and perceptually-aligned gradients.
>
> We can also help find models with better robustness-accuracy tradeoffs in the sense that we identify that it is preferable for models to be only off-manifold robust as opposed to being robust on-manifold. Thus by choosing methods that prefer only off-manifold robustness, we may be able to achieve a better accuracy-robustness tradeoff.
>
> ---
>
> 9.*“Why are the results of Imagenet 64 so low?”*
>
> On ImageNet-64x64, a performance drop is expected due to the significant downsampling of the images. We train the Resnet 18 models from scratch using standard training procedures without any significant hyperparameter optimization.

---

> > ### Comment · Reviewer_aEc6 · 2023-08-13
> > **Reviewer respone**
> >
> > I thank the authors for their response which addresses my questions and raised points.
> > As I find the paper very interesting, I am happy to raise my score.

---

### Official Review · Reviewer_avWq · 2023-07-07

**Soundness:** 2 fair
**Presentation:** 3 good
**Contribution:** 3 good
**Rating:** 6
**Confidence:** 3

**Summary:**

The paper seeks to provide a condition that leads models to have gradients that are aligned with human perception, i.e. which highlight relevant features of the image while ignoring distractor features. The paper first proposes that for any manifold, if a model is robust to perturbations which are off-manifold, then its gradients will be aligned with the manifold. The paper then defines a "signal" manifold as a hypothetical manifold of features which relevant to the classification task, defined as the multiplication of the data with a masking function. The paper then states and proves that a Bayes optimal classifier is robust to perturbations perpendicular to this manifold, and hence its gradients lie on the signal manifold. The paper further hypothesizes that gradients on the signal manifold are perceptually aligned. An argument is then provided that models robust to adversarial perturbations which have similar off-manifold robustness as the Bayes optimal classifier have perceptually aligned gradients. Finally, the paper presents empirical evidence in support of their theoretical claims and hypotheses.

**Strengths:**

1. The paper presents an interesting hypothesis and designs a robust experimental methodology to validate it within the set of assumptions.
2. The notion of weakly robust, optimally robust and overly robust models and their connection to PAGs is novel, and interesting. Comparing such classifiers with score functions is interesting in its own right, and can be a good tool in analysing models empirically.


**Weaknesses:**

1. It is not clear from the experiments what the signal manifold is. Line 278 talks about perturbing about 10\% of the input, but it would be good to clarify this further.
2. Hypothesis 1 is never validated empirically in the paper, but the experiments build upon this hypothesis by comparing the perceptual similarity with the gradients of the score function.


**Questions:**

1. How are on- and off-manifold perturbations generated exactly?
2. Is there any way to provide more empirical evidence for Hypothesis 1?

**Limitations:**

The authors address their limitations in sections 3 and 4, where they describe how much of an approximation their empirical setup and theorems are.

---

> ### Author Rebuttal · Authors · 2023-08-09
>
> Thank you for your constructive review! We address specific concerns below.
>
> 1.*“It is not clear from the experiments what the signal manifold is. Line 278 talks about perturbing about 10% of the input, but it would be good to clarify this further.”*
>
> The concept of the signal manifold arises from the observation that the input gradients of discriminate models have a tendency to highlight parts of the image that are discriminative. This should be seen in contrast to the score, which models the entire data distribution (the data manifold), including class-irrelevant background information. The signal manifold is not explicitly known. The perturbations applied in this paper are with respect to the data manifold. In Line 278 we should have been more clear. When we measure how much the model output changes in response to a change in the input, we fix the $l_2$-norm of the perturbation that is applied to the input (just as we would do with adversarial perturbations). In our experiments, the $l_2$-norm of the perturbation is about 10% of the $l_2$-norm of the input image. We will clarify this in the paper.
>
> ---
>
> 2.*“Hypothesis 1 is never validated empirically in the paper, but the experiments build upon this hypothesis by comparing the perceptual similarity with the gradients of the score function.” “Is there any way to provide more empirical evidence for Hypothesis 1?”*
>
> Our results in Figure 2 provide partial evidence for Hypothesis 1. Here we observe that the off-manifold robustness of models increases in the top row to increasing robustness, and simultaneously we observe a corresponding increase in the LPIPS metric. This shows that perceptual alignment and lying on the data manifold go hand-in-hand. We also provide results specific to the signal manifold (which lies within the data manifold) in Figure 4.
>
> ---
>
> 3.*“How are on- and off-manifold perturbations generated exactly?”*
>
> We estimate the tangent space of the data manifold using an auto-encoder. Then we draw a random normal vector and project it onto the tangent space. The part of the vector that lies in the tangent space is the on-manifold perturbation. The part of the vector that is orthogonal to the tangent space is the off-manifold perturbation. Line 257 in the main paper goes into more detail about this.

---

> > ### Comment · Reviewer_avWq · 2023-08-13
> > **Response**
> >
> > I thank the authors for their rebuttal and maintain my score recommending acceptance.

---

### Official Review · Reviewer_58Dj · 2023-07-07

**Soundness:** 4 excellent
**Presentation:** 4 excellent
**Contribution:** 4 excellent
**Rating:** 8
**Confidence:** 4

**Summary:**

This paper explores the connection between manifold alignment, perceptually-aligned gradients, and model robustness, deriving a number of novel results that explain previously observed phenomena in the explainability and robustness literatures. Crucially the paper grounds its contributions in Bayes optimal classifiers, arguing that the gradients of such classifiers are aligned with a “signal manifold”. The paper leverages denoising autoencoders to estimate the data manifold and diffusion models to estimate the gradients of a Bayes optimal classifier. Using these quantities, the paper demonstrates a number of novel results, including a strong relationship between manifold alignment and model accuracy.

**Strengths:**

This paper presents a timely and likely impactful theoretical contribution with implications not only for robust learning but also for explainability and generalization. The paper is very well organized and well-written. The introduction provides an excellent summary of the literature and current open questions. The paper disambiguates between three closely-related notions of “perceptually-aligned gradients” and provides a powerful taxonomy through the definition of Phenomena 1-3.

Drawing on prior work, the paper introduces quantitative metrics to evaluate robustness and gradient alignment with respect to a data manifold. The theoretical contributions are somewhat geometrically simplistic, but powerful. Theorem 1 has been assumed intuitively in prior work, but this paper introduces the formalisms to quantify the connection between off-manifold robustness and on-manifold gradient alignment. Argument 1-2 outline very compelling circumstantial evidence for the broader Hypotheses and will hopefully lead to future theoretical developments.

The paper provides extensive experimental evidence, including useful error bars. Figure 2 provides gorgeous plots and the correlation between manifold alignment and model accuracy is striking.

Finally, the paper makes a number of practical advances related to effective adversarial training. Figure 5 intuitively summarizes the paper’s crucial claims about three regimes of robust training.

**Weaknesses:**

* It does not seem right that the mask defining distractor vs signal features is constrained to be binary (Definition 3). This means the signal and distractor distributions are not invariant to linear transformations of $x$. Shouldn’t the only requirement be the statistical independence on line 163?
* All experiments evaluating perceptual alignment used the LPIPS metric. Wouldn’t another metric, such as cosine distance, be appropriate. This would allow for distractors that aren’t defined by a binary mask.
* Unfortunately (probably due to computational cost) different results are presented for CIFAR-10 and ImageNet, reducing the global consistency of the paper.
* The experiment shown in Figure 4 is not well explained, would it be possible to explain the setup in a little more detail using previously-established notation?

Small issues:
* The definition of Bayes optimal classifiers assumes an equal class distribution (i.e. uniform p(y)). This most likely will not be the case in most applications. Note that this does not invalidate any of the paper’s findings.
* Type on line 200 “Bayes optimal models gradients”
* Missing article on line 220: “there exists a trade-off between [the] cross-entropy loss and on-manifold robustness term”
* Missing pronoun at line 351: “meaningful in that [they] lie tangent”
* At line 354, “gradients is repeated, and do you mean the double negative “cannot be uninterpretable”?
* Figures 4 and 5 are out of order.
* The numbering of Theorems in the Supplemental material is off (the proofs continue as Theorem 3, 4 rather than 1, 2).

**Questions:**

1. Why is Theorem 2 stated with an if and only if? Don’t you prove that both statements are true for Bayes optimal classifiers?
2. Would it be possible to add an adversarially-trained model for CIFAR-10?
3. On line 228 you say “We elaborate on this argument in the Supplementary material.” Where?
4. On line 255, does “$l_2$-adersarial robust training” mean PGD applied during training?
5. In the legend for Figure 2, is “perturbation” really the right term? Isn’t the plot essentially showing the inverse of robustness? Why not use the relative robustness metric defined in Definition 1? Why is an $l_1$ divergence used here rather than the $l_2$ divergence used in Definiton 1? Can $\rho_1$ here be connected to Brier score to provide a proper score function?
6. Are error bars standard error?
7. Do you have any comment on the point where the lines cross in Figure 2 and why there is a different trade off depending on the training objective?
8. At line 274, how were these specific $\sigma$ values chosen for each dataset to parameterize the diffusion model?

**Limitations:**

The acknowledged limitations throughout the paper and the supplemental material are superb (e.g. the statement “We provide two lines of argument in support for this hypothesis [Hypothesis 2]. Ultimately, however, our evidence is empirical”).

---

> ### Author Rebuttal · Authors · 2023-08-09
>
> Thank you for your insightful review! We're glad that you liked our paper overall, including the acknowledgement of our limitations. We respond to individual questions below.
>
> 1.*“It does not seem right that the mask defining distractor vs signal features is constrained to be binary (Definition 3)”*
>
> We agree that Definition 3 can be extended such that the signal and distractor are any orthogonal complements of each other, and need not be axis aligned to the inputs. The reason we define these in this manner is to explain the observations of Shah et al. (“Do input gradients highlight discriminative features?”, NeurIPS 2021) who observe that input gradients of robust models highlight discriminative features, which they define as input pixels that can be used to predict the label. Having said that, we agree that generalizing definition 3 in this manner makes the theory potentially more interesting and general. We will add a note about this in the paper.
>
> ---
>
> 2.*“All experiments evaluating perceptual alignment used the LPIPS metric. Wouldn’t another metric, such as cosine distance, be appropriate. This would allow for distractors that aren’t defined by a binary mask.”*
>
> See our response 7 to reviewer aEc6 for the rationale behind the LPIPS metric. We are not sure that we understand the point of how the LPIPS metric does not allow for distractors that aren’t defined by a binary mask whereas the cosine distance does. We would be happy to clarify if the reviewer could elaborate on this.
>
> ---
>
> 3.*“Unfortunately (probably due to computational cost) different results are presented for CIFAR-10 and ImageNet, reducing the global consistency of the paper.” / “Would it be possible to add an adversarially-trained model for CIFAR-10?”*
>
> We partly address this by providing results for an adversarially-trained model on CIFAR-10 in the additional pdf file.
>
> ---
>
> 4.*“The experiment shown in Figure 4 is not well explained, would it be possible to explain the setup in a little more detail using previously-established notation?”*
>
> Certainly! We apologize for the lack of detail, and we will rewrite this setup accordingly. Essentially, we create a variant of the MNIST dataset where the signal and distractor are known by design. We then proceed to train robust models and find that they are indeed more robust to the distractor as opposed to the signal.
>
> ---
>
> 5.*“Why is Theorem 2 stated with an if and only if?”*
>
> The “if and only if” part of theorem 2 is indeed redundant, as we do prove that off-manifold robustness and on-manifold alignment are identical in theorem 1. We only write this again for clarity.
>
> ---
>
> 6.*“On line 228 you say “We elaborate on this argument in the Supplementary material.” Where?”*
>
> We apologize for omitting this discussion in the supplementary due to oversight. The elaboration essentially consists of a slightly more rigorous treatment of the argument presented in the main paper – we can decompose any model f(x) into on-manifold and off-manifold components globally by projecting on the tangent space at each point. Thus in the decomposition in argument 1, the on-manifold objective applies purely on the on-manifold model and similarly to the off-manifold parts. Now, if the behavior of the model is purely independent on- and off-manifold, then the stationary points of the separate objectives applied to the two independent models imply off-manifold robustness of the final model.
>
> ---
>
> 7.*“On line 255, does “adversarial robust training” mean PGD applied during training?”*
>
> Yes.
>
> ---
>
> 8.*“In the legend for Figure 2, is “perturbation” really the right term? Isn’t the plot essentially showing the inverse of robustness? Why not use the relative robustness metric defined in Definition 1? Why is an divergence used here rather than the divergence used in Definiton 1?"*
>
> Great point! The additional pdf file shows that our results are robust to the choice of divergence. We will modify Figure 2 to use the same divergence as in Definition 1. We will also exchange the term “perturbation” with “sensitivity”.
>
> Regarding plotting relative robustness vs plotting on- and off-manifold robustness separately, we plot separately to maximize clarity wrt to the underlying phenomenon. We believe relative robustness would simply scale both these curves by the overall robustness value. Nonetheless, we will add a note about this in the paper, and these to our supplement.
>
> ---
>
> 9.*“Are error bars standard error?”*
>
> The error bars in Figure 2 depict the minimum- and maximum observed values when training 10 models with different random seeds (including different regularization terms for the randomized penalties). This will be clarified in the description of the figure. The reason these are min/max instead of standard deviations is that standard deviations across 10 observations might not be very meaningful, and re-training these models is computationally expensive.
>
> ---
>
> 10.*“Do you have any comment on the point where the lines cross in Figure 2 and why there is a different trade off depending on the training objective?”*
>
> Great question! At this point, we don’t have any specific insights except for the general observation that while the overall behavior of different regularization objectives is comparable, these are still different training objectives that do lead to different optima. To given an example, while gradient norm regularization and randomized smoothing both lead to perceptually aligned gradients, a close qualitative comparison of the respective input gradients (Supplementary Figures 7 and 8) reveals that the input gradients resulting from randomized smoothing have a tendency to be perceptually more ‘smooth’.
>
> ---
>
> 11.*“At line 274, how were these specific values chosen for each dataset to parameterize the diffusion model?”*
>
> These values were chosen to maximize the perceptual quality of the estimate of the score obtained from the diffusion model.

---

> > ### Comment · Reviewer_58Dj · 2023-08-13
> > **Additional clarification**
> >
> > Thank you to the authors for a thorough and insightful rebuttal.
> >
> > I have a couple remaining questions about the experimental setup based on diffusion models.
> > 1. You say that you “choose the parameter sigma to maximize the perceptual alignment of the score”: does this mean that this parameter was based on a human visual analysis or a metrics such as LPIPS? Do you have any anecdotal evidence pertaining to the importance of the $\sigma$ parameter, e.g. confirming the need for better diffusion models to estimate the score function [R1]?
> > 2. Additionally in your response to Reviewer eaFT, you state that “The diffusion model is unconditional.” On line 271, you are aiming to estimate $\nabla_x \log p(x | y)$: this requires a class-conditional model [R2, eq 7]. What am I missing?
> > 3. I believe that this concern in (2) directly connects to my question about metrics (#2 in your rebuttal). My understanding was that your experiments with LPIPS were using the gradients of a Bayes optimal classifier provided by a conditional diffusion model. However in your response to Reviewer aEc6, you say that “A potential reason for this might be that the cosine similarity is rather sensitive to the fact that the score aligns with the data manifold, whereas model gradients align with the signal manifold.” I expected cosine similarity between the gradients of a Bayes optimal classifier and the gradients of a discriminative model to estimate precisely the _projection of the discriminative model’s gradient on the signal manifold_. Is this interpretation inconsistent with your experimental setup?
> > 4. Finally, can you confirm that this comment will be addressed in the revisions?
> > > The definition of Bayes optimal classifiers assumes an equal class distribution (i.e. uniform p(y)). This most likely will not be the case in most applications.
> >
> >
> > [R1] Wang, Zekai, Tianyu Pang, Chao Du, Min Lin, Weiwei Liu, and Y. A. N. Shuicheng. "Better Diffusion Models Further Improve Adversarial Training." ICML (2023).
> >
> > [R2] Ganz, Roy, Bahjat Kawar, and Michael Elad. "Do Perceptually Aligned Gradients Imply Robustness?." ICML (2023).

---

> > > ### Author Response · Authors · 2023-08-16
> > >
> > > Thank you for the response!
> > >
> > > *"You say that you “choose the parameter sigma to maximize the perceptual alignment of the score”: does this mean that this parameter was based on a human visual analysis or a metrics such as LPIPS? Do you have any anecdotal evidence pertaining to the importance of the $\sigma$ parameter, e.g. confirming the need for better diffusion models to estimate the score function [R1]?"*
> > >
> > > We chose the sigma parameter based on human visual analysis before running the experiments with the LPIPS metric. The ideal quantity of interest to us are the score gradients at \sigma -> 0, at which point it approximates the true noise-less score-gradients, however in practice we found these to be noisy and uninformative, thus indicating that diffusion models in practice do not recover the correct noise-less score gradients. This aligns with the observations of [R2], who also find that too small of a \sigma leads to noisy and uninformative gradients (in section 4.2, and Figure 8 of [R2]).
> > >
> > > ---
> > >
> > > *"Additionally in your response to Reviewer eaFT, you state that “The diffusion model is unconditional.” On line 271, you are aiming to estimate $\nabla_x \log p(x | y)$: this requires a class-conditional model [R2, eq 7]. What am I missing?"*
> > >
> > > We apologize for the confusion. Based on our theory, there are actually two principled ways to evaluate diffusion models and discriminative classifiers. The first is to take the input gradient with respect to a single class and compare it to a conditional diffusion model $p(x | y)$. The second is to sum the input gradients of all the classes and compare it to an unconditional diffusion model $p(x)$ (i.e., computing marginal probabilities). The results in the paper present the first approach on ImageNet and ImageNet-64x-64 and the second approach on CIFAR-10 (We clearly state this in Supplement Sections C.3 and C.4). As we indicated in our response to Reviewer eaFT, Figure 2 with a conditional diffusion model on CIFAR-10 is nearly identical (among others, because the scores of the conditional and unconditional model on natural images are very similar, which is perhaps another drawback of the current diffusion models, also observed in [R2]). The revised version will clarify these points in the main paper and also include the results with a conditional diffusion model on CIFAR-10.
> > >
> > > ---
> > >
> > > *"I believe that this concern in (2) directly connects to my question about metrics. My understanding was that your experiments with LPIPS were using the gradients of a Bayes optimal classifier provided by a conditional diffusion model. However in your response to Reviewer aEc6, you say that “A potential reason for this might be that the cosine similarity is rather sensitive to the fact that the score aligns with the data manifold, whereas model gradients align with the signal manifold.” I expected cosine similarity between the gradients of a Bayes optimal classifier and the gradients of a discriminative model to estimate precisely the projection of the discriminative model’s gradient on the signal manifold. Is this interpretation inconsistent with your experimental setup?"*
> > >
> > > We agree that the ideal quantity to use in the experiments would be the input gradients of a Bayes optimal classifier. In practice, however, we found it non-trivial to obtain these gradients from a conditional diffusion model (see Li et al., “Your Diffusion Model is Secretly a Zero-Shot Classifier”, 2023). This is why we decided to proxy the input gradients of a Bayes optimal classifier with the gradients from the score model $p(x \mid y)$ (see line 266 in the paper). The gradients of the score model lie, by construction, on the data manifold and not the signal manifold.
> > >
> > > We also agree with your intuition that the cosine similarity between the gradients of a Bayes optimal classifier and the discriminative model’s gradients can be a proxy for the projection on the signal manifold. Note, however, that the signal manifold will in general be multi-dimensional, which means that in order to project arbitrary vectors on the signal manifold we would need to perform this projection with respect to all the basis vectors of the local tangent space.
> > >
> > > ---
> > >
> > > *"Finally, can you confirm that this comment will be addressed in the revisions?"*
> > >
> > > Yes, we thank you for all the minor corrections suggested! We will make these revisions in an updated version of this draft.

---

### Official Review · Reviewer_nd2c · 2023-07-10

**Soundness:** 3 good
**Presentation:** 3 good
**Contribution:** 4 excellent
**Rating:** 6
**Confidence:** 3

**Summary:**

The paper titled "Which Models have Perceptually-Aligned Gradients? An Explanation via Off-Manifold Robustness" investigates the phenomenon of perceptually-aligned gradients (PAGs) in robust computer vision models. PAGs refer to the alignment of model gradients with human perception, enabling these models to exhibit generative capabilities such as image generation, denoising, and in-painting. The paper aims to explain the underlying mechanisms behind PAGs and provides insights into the relationship between off-manifold robustness, gradient alignment, and model accuracy.

The paper's contributions can be summarized as follows:

1. Theoretical Explanation: The paper presents a theoretical explanation of PAGs through the concept of off-manifold robustness. It demonstrates that the alignment of gradients with the data manifold is equivalent to off-manifold robustness. It also introduces the distinction between the data manifold and the signal manifold, which helps in understanding the input gradients of robust discriminative models.

2. Connection to Bayes Optimal Models: The paper establishes a connection between Bayes optimal models and off-manifold robustness. It shows that Bayes optimal models achieve both off-manifold robustness and on-manifold gradient alignment. The input gradients of Bayes optimal classifiers lie on the signal manifold, indicating their perceptual alignment.

3. Empirical Analysis: Extensive empirical analysis is conducted to validate the theoretical findings. Robust models trained with various techniques are evaluated on different datasets. The experiments confirm the relative off-manifold robustness of robust models, the correlation between off-manifold robustness and perceptual alignment, and the presence of signal-distractor decomposition in robust models.

The paper's main contribution lies in providing a theoretical framework and empirical evidence for understanding the mechanisms behind PAGs in robust computer vision models. It sheds light on the importance of off-manifold robustness and its connection to gradient alignment and model accuracy. The findings have implications for developing more explainable and generative models in computer vision tasks. By identifying different regimes of robustness, the paper also calls for rethinking standard robustness objectives and benchmarks.

Overall, the paper advances our understanding of the properties and behavior of robust models, paving the way for further research in improving model interpretability and generative capabilities in the field of computer vision.

**Strengths:**

The strength of the paper lies in its combination of theoretical analysis and empirical evaluation. It provides a comprehensive examination of the phenomenon of perceptually-aligned gradients (PAGs) in robust computer vision models. By presenting a theoretical framework based on off-manifold robustness and its connection to gradient alignment, the paper offers a solid foundation for understanding the underlying mechanisms behind PAGs.

Furthermore, the empirical analysis conducted in the paper strengthens its findings. The authors perform extensive experiments using different datasets and robust training techniques, validating their theoretical explanations and hypotheses. The empirical results consistently support the theoretical claims, providing evidence for the presence of off-manifold robustness, gradient alignment, and signal-distractor decomposition in robust models.

The paper's contribution also extends to the exploration of various aspects related to PAGs, such as the connection to Bayes optimal models, the trade-off between on- and off-manifold robustness, and the correlation between off-manifold robustness and perceptual alignment. These insights broaden our understanding of the behavior and properties of robust models in computer vision tasks.

Overall, the strength of the paper lies in its rigorous analysis, clear explanations, and compelling empirical evidence, making a significant contribution to the field of computer vision and model interpretability.

**Weaknesses:**

One potential weakness of the paper is that the evaluation of on- and off-manifold perturbations does not cover the entire space of perturbations. There is a significant portion of perturbations that fall in-between the two categories, which are not thoroughly explored in the current research. It would be valuable for future work to address these intermediate cases and provide a more comprehensive understanding of the model's behavior in those scenarios.

**Questions:**

Clarification on the choice of perturbations: Can the authors explain the rationale behind the specific choice of on- and off-manifold perturbations? How representative are these perturbations of real-world scenarios? Are there other types of perturbations that could be considered in future research?

Extension to non-linear models: The paper primarily focuses on linear models and their relationship to off-manifold robustness. Can the authors discuss how their findings and hypotheses can be extended to non-linear models? Are there any unique considerations or challenges in analyzing non-linear models in this context?



**Limitations:**

- Limited exploration of perturbation space: The choice of on- and off-manifold perturbations in the experiments might not cover the entire space of possible perturbations. There could be a wide range of perturbations that lie between the two extremes. Exploring a more comprehensive set of perturbations and their effects on on- and off-manifold robustness could enhance the depth of the analysis.
- Evaluation metrics: The paper primarily focuses on measuring robustness based on the change in model output. While this is a common approach, it may not capture all aspects of robustness. Considering additional evaluation metrics, such as adversarial examples or other robustness measures, could provide a more comprehensive understanding of model behavior.

---

> ### Author Rebuttal · Authors · 2023-08-09
>
> Thank you for your constructive and encouraging review! We are glad that you found our analyses rigorous and our explanations clear. We answer specific questions below.
>
> 1.*“One potential weakness of the paper is that the evaluation of on- and off-manifold perturbations does not cover the entire space of perturbations. There is a significant portion of perturbations that fall in-between the two categories, which are not thoroughly explored in the current research.”* *“Clarification on the choice of perturbations: Can the authors explain the rationale behind the specific choice of on- and off-manifold perturbations?”* *“Limited exploration of perturbation space: The choice of on- and off-manifold perturbations in the experiments might not cover the entire space of possible perturbations.”*
>
> Please note that by definition, every perturbation can be written as a linear combination of one on- and one off-manifold perturbation (since the tangent space is simply a linear subspace). This is also crucial to our experiments -- we create random Gaussian perturbations and then proceed to project them onto on- and off-manifold parts for analysis in Figure 2. Therefore, the results in our paper already incorporate the intermediate cases mentioned. In case we have misunderstood your questions, please let us know, we are happy to clarify.
>
> ---
>
>
> 2.*“Extension to non-linear models: The paper primarily focuses on linear models and their relationship to off-manifold robustness. Can the authors discuss how their findings and hypotheses can be extended to non-linear models? Are there any unique considerations or challenges in analyzing non-linear models in this context?”*
>
> We believe this is a misunderstanding – both the theory and the experiments in our paper are for standard deep neural networks, that is, highly non-linear models. Except for argument 2, which discusses linear models, the rest of the paper is about non-linear neural network models.
>
> ---
>
> 3.*“Evaluation metrics: The paper primarily focuses on measuring robustness based on the change in model output. While this is a common approach, it may not capture all aspects of robustness. Considering additional evaluation metrics, such as adversarial examples or other robustness measures, could provide a more comprehensive understanding of model behavior.”*
>
> We are interested in robustness insofar as it is related to the phenomenon of perceptual-aligned gradients. We provide additional results on adversarial noise in the one-page PDF, please see the global comment for details.

---

> > ### Comment · Reviewer_nd2c · 2023-08-18
> >
> > Thank you for the clarification. I have read the response, and maintain my original assessment of the paper.

---

### Author Rebuttal · Authors · 2023-08-09

We thank all reviewers for taking the time to review our paper and providing constructive feedback. We are encouraged by their positive assessment of the paper, and we are committed to incorporating reviewer feedback to further strengthen the paper.

Based on the reviewers’ comments, we have conducted additional experiments and provide the following plots in a one-page PDF:
1. Results with adversarial training on CIFAR-10 (in response to a question by reviewer 58Dj). We provide plots in the format of Figure 2 in the main paper and visualizations of input gradients as in the Supplement. We trained with projected gradient descent (PGD) and a $l_2$-budget of varying size $\epsilon$, taking 10 steps of size $\alpha = 2 * \epsilon / 10$. We used our default learning rate schedule on CIFAR-10 and proportionally decreased the initial learning rate for very large perturbation budgets (compare Supplement B.2).
2. Robustness of our evaluation with respect to different divergence measures (in response to a question by reviewer 58Dj).
3. Adding the LPIPS metric to Figure 5 (in response to reviewer aEc6).

---

### Decision · Program_Chairs · 2023-09-21

**Decision:**

Accept (spotlight)

**Comment:**

The reviewers were unanimously positive about this paper, noting that the paper addresses PAGs are an interesting but poorly understood phenomenon. This paper provides a theoretical explanation and extensive empirical validation thereof. Reviewers praised the clarity of exposition and mathematical rigour and clarity. As such I gladly recommend this paper for acceptance.